# Adaptive-Margin Masking and Restoration for Balanced Multimodal Learning

## Abstract

Real-world tasks rely on information from multiple sensory sources, motivating multimodal learning as a core paradigm in modern machine learning. However, multimodal learning suffers from modality laziness, where one modality suppresses the contribution of the other modalities by dominating the training process. Recent studies of proposing various metrics to identify lazy modalities and manipulate optimizations to mitigate this imbalance, facing two main problems: 1) The identification of lazy modality is incomplete and overly sharp; 2) The dominated modality can be lagged by under-optimized lazy modalities. To address these issues, we propose **A**daptive-Margin **M**asking and **Re**storation (**AMRe**), which introduces adaptive-margin modality identification and restoration optimization to balance dominated and lazy modalities. Experimental results show that AMRe consistently outperforms competitive baselines and several state-of-the-art methods by achieving significant improvement on standard multimodal benchmarks. The codes are released in https://anonymous.4open.science/r/AMRe-85BD.

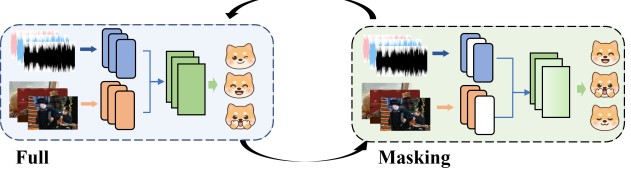

(a) Adaptive-Margin Modality Identification

(b) Restoration Strategy

*Figure 1.* (a) Adaptive-Margin Modality Identification, where left is quantified modality laziness via incorrectness and uncertainty, and right is a soft margin $\gamma$ used to distinguish dominated (white) vs. lazy (black) modality by considering ambiguous samples (gray). The objects with the same shape represent the same instance with features from different modalities. (b) Restoration alternates full and masking optimization phases.

## 1. Introduction

Models trained on unimodal data fail to solve the real-world tasks that require complementary and multisensory information (Baltrušaitis et al., 2018; Zhu et al., 2024; Yang et al., 2024). Multimodal learning, which trains models on data from diverse modalities, has emerged as a central research paradigm in modern machine learning (Sun et al., 2021a; Wei et al., 2023; Yu et al., 2023; Wu et al., 2024). However, recent multimodal learning methods struggle to effectively digest knowledge across modalities, because features from one modality across instances can dominate the

model training. As a result, modalities fail to contribute equally, leading to the problem of modality laziness (Wang et al., 2020; Huang et al., 2022; Du et al., 2023).

To address this problem, several methods have been proposed that can be broadly categorized into two directions: joint optimization and alternating optimization. Joint optimization regulates the training dynamics by tailoring learning rates (Xiao et al., 2020; Sun et al., 2021b) and gradient updates (Peng et al., 2022; Li et al., 2023; Wei et al., 2024b; 2025) based on the performance of individual modalities. Alternating optimization mitigates the suppression of lazy modalities through sample-level interventions. The effectiveness of these methods hinges on the accurate identification of the lazy modality. Representative approaches employ diverse metrics, including sample-level marginal contributions (Wei et al., 2024a) and uncertainty-based measures such as probability distribution divergence (Ma et al., 2025).

[1]Anonymous Institution, Anonymous City, Anonymous Region, Anonymous Country. Correspondence to: Anonymous Author <anon.email@domain.com>.

Preliminary work. Under review by the International Conference on Machine Learning (ICML). Do not distribute.

However, existing lazy-modality identifications are incomplete and overly-sharp. We argue that the lazy modality should be identified by *incorrectness* and *uncertainty*. As shown in the left of Figure 1 (a), incorrectness captures the modality tendency of incorrect prediction by random, while uncertainty reflects the ambiguity and unreliability of its confidence. The other problem is overly-sharp identification. As shown in the right of Figure 1 (a), the black shape of high uncertainty and incorrectness are lazy-modality samples, while the white shape are dominated-modality samples that dominate model training. The existing methods draw a hard decision boundary using various metrics to separate these samples. However, the modalities of some samples (gray shape) near the boundary are not as lazy as the black shape, and treating them as such introduces noise into the optimization. The ambiguity on laziness can mislead balanced multimodal learning.

To alleviate these problems, we propose Adaptive-Margin Masking and Restoration (AMRe) for balanced multimodal learning. We quantify the contribution of each modality by using a hybrid metric that integrates both incorrectness and uncertainty. Based on the metric, we take an *adaptive-margin modality identification* with threshold $\gamma$ to softly identify dominated and lazy modalities, as shown in the right of Figure 1 (a). During training, the dominated modality is masked to promote optimization of the lazy one. However, prolonged suppression in continuous masking blocks gradient updates, leading to feature degradation. To mitigate this effect, AMRe introduces *restoration* that periodically alternates between full and masking optimization. As shown in Figure 1 (b), this periodic reactivation enables stable co-adaptation between dominated and lazy modalities.

Experimental results on standard benchmarks of CREMA-D, AVE, MVSA, and IEMOCAP demonstrate that AMRe consistently outperforms the competitive baselines and state-of-the-art models by achieving accuracies of 80.78%, 76.37%, 75.68%, and 68.93%, respectively.

Our contributions are summarized as follows:

- We propose an adaptive-margin modality identification, which quantifies the contribution of each modality via incorrectness and uncertainty to address the challenge of modality laziness.

- We propose a unified paradigm that combines the sample-level precision of alternating optimization with the simplicity of joint training. By dynamically masking the dominated modality, AMRe decouples gradient interference without complex multi-stage pipelines.

- The experimental results show that our proposed training methods can significantly outperform baselines and existing methods by achieving the state-of-the-art results on standard benchmarks.

## 2. Related Work

### 2.1. Multimodal Fusion Mechanisms

Existing multimodal fusion methods can be categorized into parameter-free aggregation and parameterized interaction.

**Parameter-free Aggregation:** Aggregation combines multimodal features without introducing additional learnable parameters, such as concatenation(Owens & Efros, 2018) and summation. While computationally efficient, they rely on downstream classifiers to implicitly capture cross-modal correlations and thus offer limited modeling capacity.

**Parameterized Interaction:** Parameterized fusion strategies incorporate learnable modules to facilitate cross-modal interactions. FiLM (Perez et al., 2018) modulate features via conditional affine transformations. Bi-Gated (Kiela et al., 2018) apply gating mechanisms to dynamically reweight modalities. Other approaches, such as CentralNet (Vielzeuf et al., 2018) and MMTM (Joze et al., 2020), introduce centralized or channel-wise fusion modules.

These methods focus on feature integration but ignore optimization dynamics, making them vulnerable to modality laziness, where dominated modalities impede learning.

### 2.2. Balanced Multimodal Learning

Recent studies attribute modality laziness to imbalanced learning dynamics across modalities, where the dominated modality dominates optimization and leads to under-utilized multimodal information. Wang et al. (2020) identify this issue as stemming from discrepancies in convergence rates, causing multimodal models to underperform compared to unimodal baselines. Zhang et al. (2024) demonstrate that alternating the optimization of unimodal encoders can effectively mitigate the suppression imposed by the dominated modality. Based on these optimization strategies, existing methods can generally be categorized into two streams: joint optimization and alternating optimization.

**Joint Optimization** Joint optimization strategies maintain simultaneous inputs from all modalities while regulating the optimization process through objective adjustments and speed control. One line of work focuses on recalibrating optimization objectives to combat laziness (Wang et al., 2020; Fan et al., 2023; Xu et al., 2023; Du et al., 2023). A representative work is Gradient-Blending (Wang et al., 2020), which determines the optimal mixture of modality losses based on unimodal overfitting patterns. Fan et al. (2023) further addresses this by incorporating prototypical cross-entropy loss to balance inter-modal learning rates. The other line of work aligns optimization dynamics by manipulating gradients and learning rates (Sun et al., 2021b; Peng et al., 2022; Kontras et al., 2024; Wei et al., 2025).

**Alternating Optimization** Alternating optimization seeks to prevent the dominated modality from interfering with the lazy one by altering training sequences or masking multimodal inputs, thereby isolating unimodal branches to enhance their feature representation capabilities (Zhang et al., 2024; Wei et al., 2024a; Ma et al., 2025; Henriques e Silva et al., 2025). Zhang et al. (2024) decompose multimodal learning into alternating unimodal adaptation steps, preserving cross-modal interactions via a shared head. Wei et al. (2024a) introduce sample-level marginal contributions to distinguish between dominated and lazy modalities, employing resampling to boost the expressiveness of lazy modalities. Ma et al. (2025) utilize KL divergence to quantify sample uncertainty, identifying modality strength to guide a data remixing strategy for targeted training of lazy samples. AMRe bridges joint and alternating optimization by implementing sample-level gradient isolation within a joint framework. This avoids the coarse granularity of epoch-level alternation and gradient coupling of joint optimization.

# 3. Preliminary

We formulate the standard paradigm of multimodal classification, followed by identifying the prevalent issue of insufficient modality exploitation of modality laziness.

## 3.1. Multimodal Learning for Classification

We take a multimodal classification task with $M$ modalities (e.g., audio, visual, and text). $\mathcal{D} = \{(\mathbf{x}_i, y_i)\}_{i=1}^{N}$ is the dataset consisting of $N$ samples, where $\mathbf{x}_i = \{x_i^{m_j}\}_{j=1}^{M}$ is the set of heterogeneous inputs for the $i$-th sample, $m_j$ is the $j$-th modality, and $y_i \in \{1, \ldots, C\}$ is the corresponding ground-truth label. As shown in the left of Figure 2, the multimodal model adopts a dual-branch architecture comprising two unimodal encoders, $\varphi_1$ and $\varphi_2$, which are employed to extract feature representations from their corresponding modality inputs.[1] Given the input $x_i$ with two modalities $m_1$ and $m_2$, the outputs are $z_i^{m_1} = \varphi_1(x_i^{m_1}; \theta_1) \in \mathbb{R}^{d_1}$ and $z_i^{m_2} = \varphi_2(x_i^{m_2}; \theta_2) \in \mathbb{R}^{d_2}$, where $\theta_1$ and $\theta_2$ are the parameters of two unimodal encoders, respectively.

The features are concatenated to obtain a joint representation $z_i^f = [z_i^{m_1}; z_i^{m_2}]$ that is fed to a fusion classification head to generate predictions $\hat{y}_i^f = \psi_f(z_i^f) = W z_i^f + b$. The corresponding multimodal cross-entropy loss is:

$$\mathcal{L}^f = \frac{1}{|B|} \sum_{i=1}^{|B|} \mathcal{L}_{\text{CE}}(\hat{y}_i^f, y_i), \quad (1)$$

where $|B|$ denotes the batch size, and $\mathcal{L}_{\text{CE}}$ is the standard cross-entropy loss. To enforce unimodal representational

---

[1]Although shown with two modalities, the proposed framework naturally extends to three or more modalities.

capability, we introduce separate classification heads for each modality $m_j$, producing predictions $\hat{y}_i^{m_j} = \psi_j(z_i^{m_j})$. The unimodal loss for the $j$-th modality is formulated as:

$$\mathcal{L}^{m_j} = \frac{1}{|B|} \sum_{i=1}^{|B|} \mathcal{L}_{\text{CE}}(\hat{y}_i^{m_j}, y_i). \quad (2)$$

The overall training objective is defined as:

$$\mathcal{L} = \mathcal{L}^f + \alpha \cdot \sum_{m_j=1}^{M} \mathcal{L}^{m_j}, \quad (3)$$

where $\alpha$ controls the contribution of the unimodal losses.

## 3.2. The Modality Laziness Problem

During backpropagation, for a given sample $i$, the gradient of the multimodal loss with respect to the parameters $\theta_1$ can be decomposed as:

$$g_{\theta_1}^{\text{multi}} = \frac{\partial \mathcal{L}_i^f}{\partial \psi_f(z_i^f)} \cdot \frac{\partial \psi_f(z_i^f)}{\partial z_i^f} \cdot \frac{\partial z_i^f}{\partial z_i^{m_1}} \cdot \frac{\partial z_i^{m_1}}{\partial \theta_1}. \quad (4)$$

where the joint representation $z_i^f$ drives a single multimodal cross-entropy loss that propagates gradients to each unimodal branch via the shared fusion pathway. While this joint optimization integrates multimodal information, it also couples the gradients received by different unimodal branches.

To investigate the underlying cause of this imbalance, we analyze the gradient with respect to the encoder parameters $\theta_1$, formulated as $g_{\theta_1}^{\text{multi}} = \left[ \sum_{j=1}^{C} (p_{i,j} - \mathbb{I}(j = y_i)) \cdot W_j^{m_1} \right] \cdot \frac{\partial z_i^{m_1}}{\partial \theta_1}$, where $\mathbb{I}(j = y_i)$ denotes the one-hot indicator of the ground-truth class. By expanding the softmax probability $p_{i,j}$, we isolate the influence of modality $m_2$:

$$p_{i,j} = \frac{e^{W_j^{m_1} z_i^{m_1} + b_j} \cdot \rho_{j,m_2}}{\sum_{r=1}^{C} e^{W_r^{m_1} z_i^{m_1} + b_k} \cdot \rho_{r,m_2}}, \quad (5)$$

where $\rho_{r,m_2} = e^{(W_r^{m_2} - W_{y_i}^{m_2}) \cdot z_i^{m_2}}$ represents the interaction term introduced by modality $m_2$.[2] As shown in Eq. 5, the gradient of encoder $\varphi_1$ is inherently coupled with the representations $z_i^{m_2}$ and weights $W^{m_2}$ of the other modality via the softmax normalization term.

If $m_2$ is dominant, its strong representations amplify the interaction term $\rho_{r,m_2}$ in the softmax denominator. This amplification indirectly attenuates the gradient magnitude of the other modality, causing the lazy modality to receive consistently diminishing updates. To mitigate this, we eliminate cross-modal interference by setting $z_i^{m_2} = 0$, which forces the interaction term to be 1 and effectively decouples the gradients of the unimodal branches. This analysis motivates our strategy of suppressing dominated modalities during training to restore balanced optimization.

---

[2]Detailed derivation is provided in Appendix A.1.

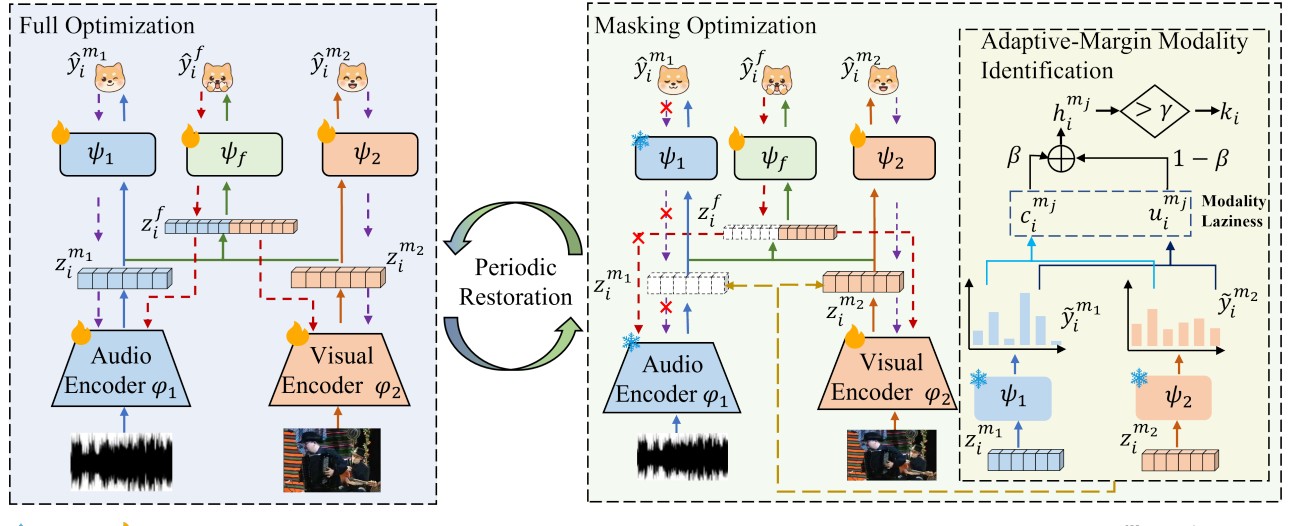

*Figure 2.* **Overview of AMRe. (Left)** Full Optimization, where all modalities are jointly optimized to maintain representation quality. **(Right)** Masking Optimization, in which the Adaptive-Margin Modality Identification module computes a laziness score from incorrectness and uncertainty to identify the dominated modality.

## 4. AMRe

We introduce the framework of Adaptive-Margin Masking and Restoration, which is shown in Figure 2. The core of the framework is adaptive-margin masking optimization. Together with full optimization, we introduce a restoration to balance modalities.

### 4.1. Full Optimization

Full optimization serves as the foundational optimization strategy. It aggregates features from all modalities $\{z_i^{m_j}\}_{j=1}^{M}$ to construct a joint representation $z_i^f$ for the final prediction. The objective is to minimize the task-specific loss (Eq. 3) using the complete multimodal input. This approach enables simultaneous gradient updates across all unimodal branches, fostering the learning of cross-modal interactions. We periodically reintegrate this full optimization as a restoration mechanism to recover unimodal feature expressiveness following the masking phases.

### 4.2. Masking Optimization

Instead of full optimization, masking optimization masks the dominated modality that is identified using adaptive-margin modality identification, while the modules corresponding to the lazy modality are trained.

**Modality Laziness** We propose a comprehensive metric to quantify the modality laziness by integrating sample-level *incorrectness* and *uncertainty* with a soft margin.

- We quantify unimodal classification performance for

*incorrectness* using the cross-entropy loss:

$$c_i^{m_j} = \mathcal{L}_{\text{CE}}(\tilde{y}_i^{m_j}, y_i), \tag{6}$$

where $\tilde{y}_i^{m_j}$ is the prediction probability obtained from the frozen classification head of modality $m_j$. A larger score $c_i^{m_j}$ indicates that modality $m_j$ provides less discriminative information for sample $x_i$ under the current training state. This suggests that $m_j$ may be under-optimized relative to other modalities and may behave as a lazy modality.

- We employ the negative Kullback–Leibler (KL) divergence to measure the deviation of the modality prediction probability distribution $\tilde{y}_i^{m_j}$ from a uniform distribution $U$ for *uncertainty*:

$$u_i^{m_j} = -D_{\text{KL}}(\tilde{y}_i^{m_j}\|U), \tag{7}$$

where $u_i^{m_j}$ quantifies the information gain provided by a modality beyond random guessing. A larger $u_i^{m_j}$ indicates the prediction approaches uniformity, indicating higher uncertainty and reduced reliability.

We formulate the laziness score by integrating incorrectness and uncertainty:

$$h_i^{m_j} = \beta \cdot \sigma_\tau(c_i^{m_j}) + (1 - \beta) \cdot \sigma_\tau(u_i^{m_j}), \tag{8}$$

where $\sigma_\tau(\cdot)$ is the temperature-scaled softmax normalization, which projects the raw metrics into a normalized probability distribution to ensure commensurability across modalities. We set the temperature $\tau = 0.2$ to control the sharpness of the score distribution. The coefficient $\beta \in [0, 1]$ governs the trade-off between incorrectness and uncertainty.

**Adaptive-Margin Modality Identification** For sample $x_i$, $\mathcal{H}_i = \{h_i^1, \ldots, h_i^M\}$ is the set of laziness scores across all modalities. We quantify the optimization imbalance by computing the laziness scores gap $\Delta_i$:

$$\Delta_i = \max(\mathcal{H}_i) - \min(\mathcal{H}_i), \tag{9}$$

We introduce a soft margin $\gamma$ to filter out fuzzy samples. The adaptive mask index $k_i$ is formally defined as:

$$k_i = \begin{cases} m_i^*, & \text{if } \Delta_i \geq \gamma, \\ 0, & \text{otherwise.} \end{cases} \tag{10}$$

where $m_i^* = \arg\min(\mathcal{H}_i)$ denotes the dominated modality index with the lowest laziness score. $m_i^*$ is selected to be masked by setting $k_i = m_i^*$ if $\Delta_i \geq \gamma$, and modality is kept for optimization by setting $k_i = 0$ if $\Delta_i < \gamma$.

**Fusion Feature with Masked Modality** We apply the masking operation to the feature embeddings before fusion. If modality $m_j$ is selected to be masked (i.e., $k_i = m_j$), its feature vector $z_i^{m_j}$ is replaced by a zero vector $\mathbf{0}$; otherwise, the original feature is retained. The joint representation $z_i^f$ is obtained by concatenating these mask-conditioned features. This zero-masking operation explicitly blocks the gradient flow from the fusion loss to the dominated encoder, enforcing the optimization of the remaining modalities.

### 4.3. Training Strategy

We propose a restoration that periodically reverts to full optimization, which is shown in Figure 2. Unlike continuous masking where prolonged suppression leads to optimization lag, this strategy prevents representation degradation by regularly enabling full gradient updates. Algorithm 1 details the AMRe pipeline. After warm-up epochs, the optimization mode alternates between full and masking optimization via the indicator $I_{\text{mask}}$, regulated by the restoration interval $E_p$.

In adaptive-margin masking, the standard unimodal loss formulation becomes inadequate, as samples with masked modalities should not propagate gradients through their corresponding unimodal branches. To address this limitation, we reformulate the unimodal loss of Eq. 2 for modality $m_j$:

$$\mathcal{L}^{m_j} = \frac{\sum_{i=1}^{|B|}(\mathbb{I}(k_i \neq m_j)\,\mathcal{L}_{\text{CE}}(\hat{y}_i^{m_j}, y_i))}{\sum_{i=1}^{|B|} \mathbb{I}(k_i \neq m_j)}, \tag{11}$$

where $k_i \in \{0, 1, \ldots, M\}$ denotes the modality masked for sample $i$, and $\mathbb{I}(\cdot)$ is the indicator function. $\mathcal{L}^{m_j}$ reduces to the standard unimodal loss of Eq. 2 if no modality is masked. Together with the standard fusion loss obtained via Eq. 1, the overall training objective of AMRe is defined as:

$$\mathcal{L} = \mathcal{L}^f + \alpha \cdot \sum_{m_j=1}^{M} \mathcal{L}^{m_j}, \tag{12}$$

---

**Algorithm 1** AMRe Training Strategy

---

**Inputs:** dataset $\mathcal{D} = \{(x_i, y_i)\}_{i=1}^N$, encoders $\varphi_{m_j}$, fusion head $\psi_m$, unimodal head $\psi_j$ for $j$-th modality, restoration interval $E_p$, training epoch $E$.

**for** $e = 1$ **to** $E$ **do**
    // *Condition*
    $I_{\text{mask}} \leftarrow (e \mod E_p \neq 1)$
    **for** each mini-batch $\mathcal{B} \subset \mathcal{D}$ **do**
        Extract features $z^{m_j}$ for all $j \in \{1..M\}$.
        **if** $I_{\text{mask}}$ **then**
            // *Adaptive-Margin Masking*
            Obtain temporary predictions $\tilde{y}^{m_j}$.
            Compute Incorrectness $c_i^{m_j}$ ( Eq. 6)
            Compute Uncertainty $u_i^{m_j}$ (Eq. 7).
            Compute Laziness Score $h_i^{m_j}$ ( Eq. 8).
            Obtain adaptive mask index $k_i$ (Eq. 10).
        **else**
            // *Restoration*
            Set $k_i \leftarrow 0$ for all samples.
        **end if**
        Obtain the joint representation $z^f$ according to $k$.
        Obtain $\hat{y}^f \leftarrow \psi_f(z^f)$, $\quad \hat{y}^{m_j} \leftarrow \psi_j(z^{m_j})$.
        Update the model according to the Loss (Eq. 12).
    **end for**
**end for**

---

which jointly optimizes the multimodal fusion objective and the mask-conditioned unimodal objectives.

## 5. Experiments

Experiments are carried out on standard multimodal benchmarks, and we compare the proposed AMRe to several competitive baselines.

### 5.1. Experimental Setup

**Datasets** The benchmarks cover diverse modalities: CREMA-D (Cao et al., 2014) and AVE (Tian et al., 2018) for audio-visual tasks, MVSA-Single (Niu et al., 2016) for visual-text tasks, and IEMOCAP (Busso et al., 2008) for audio-visual-text tasks.[3]

**Baselines** We group existing approaches into three categories. (1) *Conventional fusion* methods include parameter-free aggregation (Owens & Efros, 2018), such as concatenation (Concat) and summation (Sum), as well as parameterized interaction mechanisms, including FiLM (Perez et al., 2018) and Bi-Gated (Kiela et al., 2018); (2) *Joint optimization* methods include OGM and OGM-GE (Peng et al., 2022), as well as LFM (Yang et al., 2024); (3) *Alternating*

---

[3]Detailed dataset statistics are provided in Appendix C.1

*Table 1.* Results(%) across baselines with our proposed methods (AMRe). **Bold** is the best result, and underlining is the second best. * is parameterized interaction.

| Method | CREMA-D | AVE | MVSA | IEMOCAP | Avg |
|---|---|---|---|---|---|
| *Baseline Models* | | | | | |
| Concat | 73.52 | 72.89 | 73.18 | 65.38 | 71.24 |
| Sum | 73.66 | 70.40 | 73.39 | 69.38 | 71.71 |
| *FiLM | 71.51 | 68.66 | 72.35 | 66.72 | 69.81 |
| *Bi-Gated | 71.91 | 71.89 | 72.73 | 66.72 | 70.81 |
| *With Our Method* | | | | | |
| Concat + AMRe | **80.78** | **76.37** | **75.68** | 68.93 | **75.44**(+4.20) |
| Sum + AMRe | 76.08 | 71.14 | 73.60 | **71.30** | 73.03(+1.32) |
| *FiLM + AMRe | 73.39 | 70.40 | 73.80 | 67.31 | 71.23(+1.42) |
| *Bi-Gated + AMRe | 75.13 | 72.39 | 73.60 | 67.90 | 72.25(+1.44) |

*optimization* approaches include MLA (Zhang et al., 2024), Resample (Wei et al., 2024a), Remix (Ma et al., 2025), and AMST (Henriques e Silva et al., 2025).[4]

**Backbones**  ResNet-18 (He et al., 2016) for audio-visual, BERT (Devlin et al., 2019) + MAE (He et al., 2022) for visual-text, and a combination of BERT (text encoder) and ResNet-18 (audio-visual encoder) for tri-modal tasks.[5]

### 5.2. Results

**AMRe with Conventional Fusion**  To evaluate the universality and plug-and-play nature of AMRe, we first verify its effectiveness within parameter-free aggregation paradigms. As shown in Table 1, although these methods are computationally efficient, they lack intrinsic mechanisms for balanced optimization. Both parameter-free aggregation baselines obtain significant performance gains if integrated with AMRe, by achieving the improvements of 4.20% and 1.32%, respectively. Additionally, AMRe benefits parameterized interaction by achieving 1.43% on average. This suggests that AMRe effectively alleviates modality imbalance arising from simple fusion operations. We adopt Concat as the baseline fusion strategy in subsequent experiments. Parameterized fusion models underperform parameter-free aggregation in certain cases. Detailed analysis is provided in Appendix B.1.

**Comparison with SOTA models**  As shown in Table 2, AMRe is evaluated against a broad set of state-of-the-art baselines and consistently achieves competitive performance across all benchmarks. AMRe outperform the optimization-based approaches such as LFM, highlighting the advantage of adaptive-margin masking over global gradient modulation. Additionally, AMRe surpasses recent alternating strategies, including AMST and MLA, indicating that adaptive-margin modality identification combined with periodic restoration is more effective than rigid alternating schemes for mitigating modality laziness. While

---

[4]Further details are available in Appendix C.2.

[5]Implementation details can be found in Appendix C.3.

AMST adopts a skipping strategy to reduce modality dominance, its control remains coarse-grained, relying on a fixed epoch-level freezing rather than sample-adaptive regulation. Moreover, prolonged freezing of dominated modality can lead to degraded feature representations over time. By contrast, the restoration mechanism in AMRe periodically reactivates joint optimization in a sample-adaptive manner, which helps preserve feature quality throughout training. As a result, AMRe attains the highest average accuracy of 75.44%, demonstrating robust performance in balanced multimodal learning.

**Training Time**  In the same experimental setup on a single V100 GPU, AMRe introduces less than 1.5% additional latency compared to the Concat baseline. This efficiency is mainly due to the lightweight metric computations ($O(C)$) relative to backbone feature extraction, and the fact that masking is skipped during the restoration phases. Detailed runtime and FLOPs analysis are provided in Appendix D.1.

## 6. Analysis

We provide insight analysis to elaborate the effect of AMRe.

### 6.1. Restoration

We conduct a systematic analysis of the restoration, as shown in Figure 3, focusing on representation stability, masking frequency, and capability upper bound.

**Stability**  As shown in Figure 3(a), unimodal gradient similarity over time highlights the issue of feature degradation under different optimization strategies. Concat shows a clear decline in training stages, indicating an increasing optimization imbalance as modality learning diverges from the joint objective. AMRe w/o restoration mitigates this sharp decline but converges to a relatively low gradient similarity, suggesting that prolonged suppression by masking dominated modality limits the alignment between modalities. AMRe maintains consistently high gradient similarity throughout training. By periodically reintroducing full optimization, it stabilizes gradient interactions and alignments.

**Masking Frequency**  Figure 3(b) shows the frequency of masking dominated modality across different strategies. AMRe w/o restoration demonstrate the large gap between the two modality, where the Audio modality is masked far more frequently than the Visual modality, indicating excessive suppression of the stronger modality. However, equipped with restoration, AMRe enables the suppressed modality to re-align with the joint objective, effectively breaking the over-suppression loop and yielding a more stable and balanced masking distribution.

*Table 2.* Results of accuracy (%) and training time (second) with our models and SOTA models. **Bold** is the best accuracy, and underlining is the second best.

| Dataset | Metric | Method | | | | | | | | | |
|---|---|---|---|---|---|---|---|---|---|---|---|
| | | Concat | OGM | OGM-GE | LFM | MLA | Resample | Remix | AMST$_{Joint}$ | AMST$_{FULL}$ | **AMRe** |
| CREMA-D | Acc | 73.52 | 74.06 | 73.45 | 77.28 | 76.48 | 75.00 | 74.52 | 78.20 | 80.50 | **80.78** |
| | Time | 21.24 | 22.02 | 22.25 | 21.45 | 20.37 | 35.93 | 25.87 | 18.24 | 34.66 | 21.45 |
| AVE | Acc | 72.89 | 72.89 | 72.64 | 72.39 | 75.62 | 71.89 | 72.69 | 72.79 | 74.38 | **76.37** |
| | Time | 17.15 | 17.63 | 17.94 | 17.17 | 16.78 | 22.92 | 21.60 | 15.43 | 30.93 | 17.39 |
| MVSA | Acc | 73.18 | 72.97 | 74.22 | 73.31 | 75.26 | 74.43 | 74.01 | 72.50 | 72.60 | **75.68** |
| | Time | 36.64 | 37.12 | 37.63 | 36.81 | 36.28 | 56.32 | 38.52 | 33.76 | 64.65 | 36.70 |
| IEMOCAP | Acc | 65.38 | 65.68 | 65.79 | 66.42 | 67.01 | 66.57 | 67.16 | 67.76 | 68.40 | **68.93** |
| | Time | 38.73 | 39.99 | 40.16 | 38.94 | 38.64 | 66.50 | 40.95 | 36.58 | 85.41 | 39.04 |
| Avg | Acc | 71.24 | 71.40 | 71.52 | 72.52 | 73.59 | 71.98 | 72.10 | 72.81 | 73.97 | **75.44** |

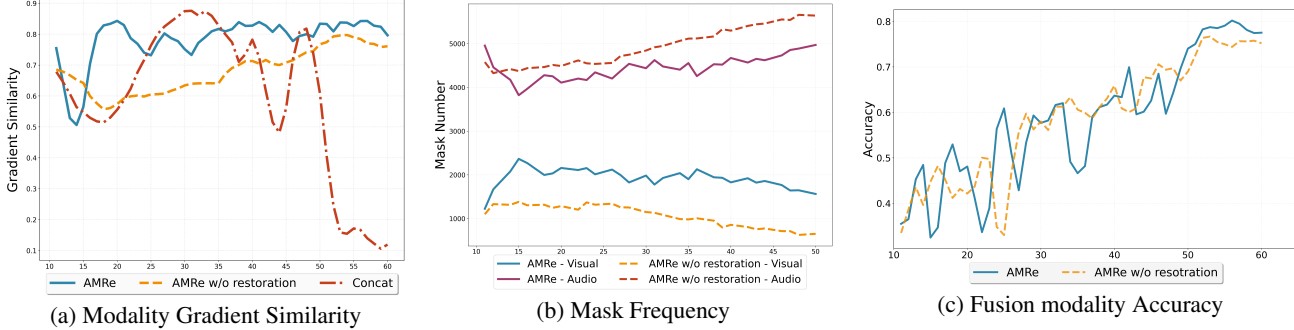

(a) Modality Gradient Similarity  (b) Mask Frequency  (c) Fusion modality Accuracy

*Figure 3.* Visualization of dynamics optimization across different strategies, where the horizontal axis represents training epochs.

**Capability Upper Bound**  As shown in Figure 3(c), AMRe exhibits smoother convergence with higher final accuracies comparing to that without restoration. Although masking optimization can address the laziness of the modality, it weakens the dominated modality. Restoration recovers the contribution of dominated modality by alternative training between full and marking optimizations. Although the training curves of AMRe are tortuous. it jumps to the convergence with better performance.

### 6.2. Ablation Study

Table 3 presents a comprehensive ablation study to examine the contributions of incorrectness , uncertainty, soft margin, and restoration interval.

**Restoration Interval**  Comparing the variant without restoration to the full model underscores the importance of periodic restoration. When restoration is disabled, continuous masking keeps one modality suppressed for extended periods, which can lead to representational drift and degraded features. This effect is most pronounced on AVE, where performance even falls below the baseline. Reintroducing the restoration phase periodically resumes full optimization, counteracting this degradation and yielding

consistent improvements.

**Incorrectness versus Uncertainty**  Uncertainty generally yields higher performance than incorrectness alone, suggesting that prediction confidence can provide a more informative signal than loss magnitude for identifying lazy modalities. However, neither cue is sufficient by itself. Combining uncertainty and incorrectness consistently yields more stable performance, highlighting their synergy in accurately characterizing laziness. This integration builds a holistic view of learning dynamics, ensuring robustness against data noise and calibration errors.

**Soft Margin**  This strategy yields consistent improvements, particularly on ambiguity-heavy datasets like IEMO-CAP, where performance increases by 2.51% compared to the hard masking baseline. By adopting a soft margin to identify the dominated modality, the model can avoid rigid decision boundaries when modality contributions are comparable. This helps reduce unnecessary masking oscillations caused by noise and stabilizes the optimization process.

Enabling all four components delivers the most balanced performance. While certain partial variants can match or slightly exceed the full model on a single dataset, they tend

*Table 3.* Ablation study on AMRe. We investigate the impact of incorrectness ($c$), uncertainty ($u$), the soft margin ($\gamma$) and the restoration interval($E_p$). The symbol ✓ indicates the inclusion of the corresponding component.

| $c$ | $u$ | $\gamma$ | $E_\mathrm{p}$ | CREMA-D | AVE | MVSA | IEMOCAP |
|---|---|---|---|---|---|---|---|
| ✗ | ✗ | ✗ | ✗ | 73.52 | 72.89 | 73.18 | 65.38 |
| ✓ | ✓ | ✓ | ✗ | 77.55 | 71.80 | 73.39 | 67.90 |
| ✓ | ✗ | ✗ | ✓ | 79.84 | 72.64 | 73.18 | 68.34 |
| ✗ | ✓ | ✗ | ✓ | 75.13 | 75.16 | 73.80 | 67.01 |
| ✓ | ✓ | ✗ | ✓ | 78.76 | 75.37 | 74.43 | 66.42 |
| ✓ | ✗ | ✓ | ✓ | **80.78** | 73.63 | 73.60 | 67.31 |
| ✗ | ✓ | ✓ | ✓ | 75.94 | 75.12 | 74.01 | 65.83 |
| ✓ | ✓ | ✓ | ✓ | 78.36 | **76.37** | **75.68** | **68.93** |

to underperform elsewhere, whereas the complete configuration remains consistently strong across benchmarks.

### 6.3. Hyperparameters

As shown in Table 4, we investigate the effect of key hyperparameters of coefficient $\beta$ to balance incorrectness and uncertainty for laziness identification, soft margin $\gamma$ and restoration interval $E_p$.[6]

$\beta$ CREMA-D tends to use uncertainty to identify modality laziness ($\beta = 0$), while IEMOCAP, AVE and MVSA favor incorrectness, where incorrectness provides a clearer signal of optimization imbalance than uncertainty. This suggests that the choice of $\beta$ is dataset-dependent and critical for effectively identifying lazy modalities.

$\gamma$ The performance of AMRe can be impreoved by adjusting $\gamma$. Models with soft margin to identify the modality laziness can outperform that with sharp identification ($\gamma = 0$). In particular, CREMA-D and IEMOCAP benefit from looser margins, which stabilize masking near decision boundaries, while AVE and MVSA favor tighter margins. It suggests that the margin $\gamma$ enables the robustness of the laziness identification, contributing to the balanced learning.

$E_\mathrm{p}$ The best performances fall within $E_\mathrm{p} \in [2, 4]$, suggesting that periodic restoration avoids the dominated modality to be lagged by the over-training lazy modality if we apply masking optimization. However larger intervals, i.e. masking optimization dominates the training, result in clear drop of performance, suggesting that prolonged suppression causes irreversible feature degradation.

### 6.4. Robustness Analysis

To evaluate robustness across different models, we evaluate all methods on IEMOCAP dataset with various missing modality rates (Zhang et al., 2024). As shown in Table 5,

[6]Full results can be found in Appendix D.2.

*Table 4.* Results (%) of the sensitivity analysis with respect to hyperparameters $\beta$, $\gamma$, and $E_p$. **Bold** indicates the best results.

| Setting | | CREMA-D | AVE | MVSA | IEMOCAP |
|---|---|---|---|---|---|
| $\beta$ | 0 | **80.78** | 73.63 | 73.60 | 67.31 |
| | 0.3 | 78.36 | 71.89 | 73.18 | 67.46 |
| | 0.7 | 76.74 | **76.37** | 75.47 | **68.93** |
| | 0.9 | 74.60 | 75.37 | **75.68** | 65.53 |
| $\gamma$ | 0 | 79.84 | 75.37 | 74.43 | 66.42 |
| | 0.05 | 78.90 | **76.37** | 74.22 | 67.46 |
| | 0.10 | 78.09 | 75.87 | **75.68** | 65.38 |
| | 0.15 | **80.78** | 75.87 | 73.60 | **68.93** |
| $E_p$ | 1 | 77.55 | 71.80 | 73.39 | 67.90 |
| | 2 | 80.65 | **76.37** | **75.68** | 65.24 |
| | 3 | 80.65 | 73.13 | 73.60 | **68.93** |
| | 4 | **80.78** | 75.62 | 74.22 | 65.58 |
| | 5 | 79.30 | 73.88 | 73.80 | 67.16 |

*Table 5.* Results (%) of different missing modality rates on the IEMOCAP dataset. **Bold** indicates the best results.

| Method | 0 | 10% | 20% | 30% | 40% | 50% |
|---|---|---|---|---|---|---|
| Concat | 65.38 | 61.98 | 57.99 | 53.70 | 51.04 | 45.86 |
| OGM | 65.98 | 62.28 | 58.43 | 55.18 | 52.12 | 46.60 |
| MLA | 67.01 | 62.42 | 58.73 | 54.69 | 52.22 | 46.01 |
| Resample | 66.80 | 62.87 | 58.73 | 55.33 | 52.51 | 45.97 |
| Remix | 65.24 | 63.46 | 59.76 | 56.66 | 53.99 | 47.78 |
| AMST$_\mathrm{full}$ | 68.40 | 63.43 | 60.73 | 56.89 | 54.12 | 47.62 |
| AMRe | **68.90** | **63.90** | **61.83** | **57.36** | **54.51** | **48.04** |

AMRe achieves the best performance across missing rates from 10% to 50%, In particular, compared with the baseline Remix, AMRe consistently yields higher accuracy, with a notable improvement of 2.07% at the 20% missing rate. demonstrating the robustness of AMRe.

## 7. Conclusion

We present Adaptive-Margin Masking and Restoration, a novel approach to address the challenge of modality laziness for balanced multimodal learning. AMRe proposes to identify laziness of modality with a soft margin, considering incorrectness and uncertainty of modality. Based on identification, we effectively adopt the feature of lazy modalities by masking the dominant modality to propose masking optimization. Additionally, together with restoration, AMRe balances the multimodal training avoiding the masked dominated modality to be lagged by the over-training of lazy modality. The experimental results with insight analysis show that the proposed model outperforms the state-of-the-art models on different benchmarks.

## Impact Statement

This work studies optimization strategies for balanced multimodal learning. While our contribution is primarily methodological, experiments are conducted on multimodal benchmarks involving affective signals (e.g., CREMA-D and IEMOCAP). Improving modality balance may enhance the robustness of multimodal models in settings where certain modalities are noisy or unreliable, potentially benefiting accessibility-related applications. As with other work involving affective data, there is a risk of misuse in surveillance or manipulative systems; however, this study focuses on the optimization of general dynamic training frameworks rather than application-specific deployments.

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

# A. Appendix

## A.1. Theoretical Derivation of Parameter-free Aggregation

In this section, we provide the detailed mathematical derivation regarding the gradient coupling issue discussed in Section 3.2. We analyze how the representation of one modality affects the gradient updates of the other and show how our masking strategy effectively decouples this interaction. Consider a multimodal classification task with two modalities $m_1$ and $m_2$. Let $z_i^{m_1} \in \mathbb{R}^{d_1}$ and $z_i^{m_2} \in \mathbb{R}^{d_2}$ denote the feature representations for sample $i$. In the standard concatenation fusion, the joint representation is $z_i^f = [z_i^{m_1}; z_i^{m_2}]$. The prediction logits for class $j$ are given by:

$$s_{i,j} = W_j z_i^f + b_j = W_j^{m_1} z_i^{m_1} + W_j^{m_2} z_i^{m_2} + b_j, \tag{13}$$

where $W_j = [W_j^{m_1}; W_j^{m_2}]$ represents the classifier weights corresponding to each modality.

**Gradient Coupling**    The cross-entropy loss for sample $i$ with ground-truth label $y_i$ is $\mathcal{L}^f = -\log(\frac{e^{s_{i,y_i}}}{\sum_k e^{s_{i,k}}})$. The gradient with respect to the encoder parameters $\theta_1$ of modality $m_1$ depends on the gradient $z_i^{m_1}$:

$$\frac{\partial \mathcal{L}^f}{\partial z_i^{m_1}} = \sum_{j=1}^{C} (p_{i,j} - \mathbb{I}(j = y_i)) \cdot W_j^{m_1}, \tag{14}$$

where $p_{i,j} = \text{softmax}(s_i)_j$. To explicitly see the influence of modality $m_2$, we expand the softmax probability $p_{i,j}$. By dividing the numerator and denominator by $e^{W_{y_i}^{m_2} z_i^{m_2}}$, we can isolate the contribution of $m_2$ into an interaction term:

$$p_{i,j} = \frac{e^{s_{i,j}}}{\sum_r e^{s_{i,r}}} = \frac{e^{W_j^{m_1} z_i^{m_1} + b_j} \cdot \rho_{j,m_2}}{\sum_r e^{W_r^{m_1} z_i^{m_1} + b_r} \cdot \rho_{r,m_2}}, \tag{15}$$

where $\rho_{r,m_2} = e^{(W_r^{m_2} - W_{y_i}^{m_2}) \cdot z_i^{m_2}}$ is the interaction term introduced by modality $m_2$. If modality $m_2$ is dominant (i.e., it has high confidence and large feature magnitudes), the term $\rho_{y_i,m_2}$ (where $r = y_i$) becomes significantly larger than other terms $\rho_{r,m_2}$ (where $r \neq y_i$). This dominance in the denominator suppresses the relative contribution of the $m_1$ terms, thereby shrinking the gradient magnitude passed to $m_1$.

**Decoupling via Masking**    In our proposed masking strategy, we temporarily set the features of the dominated modality to zero: $z_i^{m_2} = \mathbf{0}$. Substituting this into the interaction term:

$$\rho_{r,m_2} = e^{(W_r^{m_2} - W_{y_i}^{m_2}) \cdot \mathbf{0}} = e^0 = 1, \quad \forall r \in \{1, \ldots, C\}. \tag{16}$$

Consequently, the gradient calculation for $m_1$ becomes:

$$\hat{p}_{i,j} = \frac{e^{W_j^{m_1} z_i^{m_1} + b_j} \cdot 1}{\sum_r e^{W_r^{m_1} z_i^{m_1} + b_r} \cdot 1}. \tag{17}$$

As shown, the interaction term $\rho$ becomes a constant 1, completely removing the suppressive influence of $W^{m_2}$ and $z_i^{m_2}$ from the gradient path of $m_1$. This forces the optimizer to update $\theta_1$ solely based on the discriminative power of modality $m_1$, thus restoring the optimization balance.

# B. Analysis of Parameterized Interaction

## B.1. Theoretical Derivation

As shown in Figure 4, the multimodal model adopts a dual-branch architecture comprising two unimodal encoders, $\varphi_1$ and $\varphi_2$, employed to extract feature representations from their corresponding modality inputs. The outputs are $z_i^{m_1} = \varphi_1(x_i^{m_1}; \theta_1) \in \mathbb{R}^{d_1}$ and $z_i^{m_2} = \varphi_2(x_i^{m_2}; \theta_2) \in \mathbb{R}^{d_2}$, where $\theta_1$ and $\theta_2$ are the parameters of the unimodal encoders. We introduce specific unimodal refinement modules $\psi_{\tau_1}, \psi_{\tau_2}$ and a shared fusion module $\psi_{\tau_m}$. The corresponding outputs are formulated as:

$$z_i^{r_1} = \psi_{\tau_1}(z_i^{m_1}; \theta_{\tau_1}) \in \mathbb{R}^{d_1}, \quad z_i^{r_2} = \psi_{\tau_2}(z_i^{m_2}; \theta_{\tau_2}) \in \mathbb{R}^{d_2} \tag{18}$$

*Figure 4.* **Parameterized fusion optimization.** Unlike parameter-free aggregation, learnable fusion modules ($\psi_{\tau_m}$) interact with unimodal features before prediction, introducing additional gradient coupling between modalities. Dashed lines denote gradient propagation paths.

$$z_i^{\tau_{m_1}}, z_i^{\tau_{m_2}} = \psi_{\tau_m}(z_i^{m_1}, z_i^{m_2}; \theta_{\tau_m}) \tag{19}$$

These feature components are concatenated to form the joint representation $z_i^f = [z_i^{r_1}; z_i^{\tau_{m_1}}; z_i^{\tau_{m_2}}; z_i^{r_2}]$, where $[;]$ is the concatenation operation. Based on $z_i^f$, the prediction function is defined as:

$$\psi_f(z_i^f) = W z_i^f + b \tag{20}$$

The corresponding multimodal cross-entropy loss is defined as:

$$\mathcal{L}^f = \mathcal{L}_{CE}(\psi_f(z_i^f), y_i) \tag{21}$$

During backpropagation, the gradient of the loss with respect to the encoder parameters of the second modality, $\theta_2$, can be decomposed using the chain rule:

$$g_{\theta_2}^{multi} = \frac{\partial \mathcal{L}^f}{\partial \psi_f(z_i^f)} \cdot \frac{\partial \psi_f(z_i^f)}{\partial z_i^f} \cdot \left( \underbrace{\frac{\partial z_i^f}{\partial z_i^{r_2}} \cdot \frac{\partial z_i^{r_2}}{\partial z_i^{m_2}}}_{\text{Direct Path}} + \underbrace{\frac{\partial z_i^f}{\partial z_i^{\tau_{m_1}}} \cdot \frac{\partial z_i^{\tau_{m_1}}}{\partial z_i^{m_2}} + \frac{\partial z_i^f}{\partial z_i^{\tau_{m_2}}} \cdot \frac{\partial z_i^{\tau_{m_2}}}{\partial z_i^{m_2}}}_{\text{Coupled Fusion Path}} \right) \cdot \frac{\partial z_i^{m_2}}{\partial \theta_2} \tag{22}$$

$$= (\text{softmax}(\psi_f(z_i^f)) - y) \cdot W \cdot (\psi'_{\tau_2}(z_i^{r_2}) + \psi'^1_{\tau_m}(z_i^{m_1}, z_i^{m_2}) + \psi'^2_{\tau_m}(z_i^{m_1}, z_i^{m_2})) \cdot \frac{\partial z_i^{m_2}}{\partial \theta_2}. \tag{23}$$

This derivation reveals that the gradient update for modality $m_2$ is influenced by modality $m_1$ through two distinct pathways: the joint softmax calculation and the parameterized interaction module. Consequently, parameterized interaction inherently exhibits stronger gradient coupling than non-parameterized approaches, which explains its generally inferior performance when optimization imbalances occurs.

We analyze why AMRe yields smaller performance gains on parameterized interaction compared to parameter-free aggregation. Assuming $m_1$ is identified as the dominated modality, we apply masking such that $z_i^{m_1} = \mathbf{0}$. Eq. 19 then becomes:

$$z_i^{\tau'_{m_1}}, z_i^{\tau'_{m_2}} = \psi_{\tau_m}(\mathbf{0}, z_i^{m_2}; \theta_{\tau_m}) \tag{24}$$

The joint feature is updated to $z_i^f = [\mathbf{0}; z_i^{\tau'_{m_1}}; z_i^{\tau'_{m_2}}; z_i^{r_2}]$. Accordingly, the gradient in Eq. 23 is updated as:

$$g_{\theta_2}^{multi} = (\text{softmax}(\psi_f(z_i^f)) - y)) \cdot (\psi'_{\tau_2}(z_i^{r_2}) + \psi'^1_{\tau_m}(\mathbf{0}, z_i^{m_2}) + \psi'^2_{\tau_m}(\mathbf{0}, z_i^{m_2})) \cdot \frac{\partial z_i^{m_2}}{\partial \theta_2}. \tag{25}$$

*Table 6.* Results(%) of performance of AMRe combined with different fusion mechanisms. **Bold** indicates the best results.

| Dataset | Concat | | MMTM | | CentralNet | | CAEF | |
|---|---|---|---|---|---|---|---|---|
| | Base | + AMRe | Base | + AMRe | Base | + AMRe | Base | + AMRe |
| CREMA-D | 73.52 | **80.78** | 66.40 | **68.68** | 66.67 | **70.97** | 68.95 | **76.48** |
| AVE | 72.89 | **76.37** | 65.67 | **68.41** | 69.40 | **73.13** | 69.90 | **73.13** |

Although masking $z_i^{m_1}$ eliminates its direct contribution to the joint representation, the gradient flow for $m_2$ remains entangled with the fusion module $\psi_{\tau_m}$. Unlike parameter-free aggregation, where masking completely decouples the modalities, parameterized interaction retains implicit coupling through the learnable parameters of $\psi_{\tau_m}$. Even with $z_i^{m_1} = 0$, the fusion module's gradients ($\psi_{\tau_m}'^1$ and $\psi_{\tau_m}'^2$) still participate in backpropagation, exerting an implicit influence on $m_2$. This residual coupling dampens the effectiveness of the explicit masking strategy, explaining why the performance boost of AMRe on parameterized interaction is observable but less pronounced than on parameter-free architectures.

### B.2. Empirical Verification

To empirically verify these theoretical insights, we evaluate AMRe on three representative parameterized interaction models: MMTM (Joze et al., 2020), CentralNet (Vielzeuf et al., 2018), and CAEF (C.2). The results in Table 6 demonstrate that AMRe consistently enhances performance across diverse fusion paradigms, confirming its robustness. However, the observation that parameterized methods lag behind simple concatenation aligns with our derivation, suggesting that their inherent gradient entanglement intensifies modality imbalance and hinders the optimization of unimodal features.

## C. Experiment Description

### C.1. Dataset

To ensure a fair comparison and align with established protocols in prior studies (Zhang et al., 2024; Henriques e Silva et al., 2025), we evaluate the effectiveness of our proposed AMRe framework on four representative multimodal datasets. Most datasets are partitioned into training, validation, and testing sets with an approximate ratio of 8:1:1, except for CREMA-D, which follows a 9:1 training-testing split without a validation set, consistent with prior works. The specific statistics and partition details are summarized in Table 7. Detailed descriptions of each dataset are provided below:

**CREMA-D (Cao et al., 2014)**   Designed for multimodal emotion recognition, CREMA-D consists of facial and vocal emotional expressions. It contains 7,442 original clips featuring actors from diverse backgrounds enacting various emotional states. The dataset covers six universal emotion categories: *Anger, Disgust, Fear, Happiness, Neutrality,* and *Sadness.*

**AVE (Audio-Visual Event) (Tian et al., 2018)**   The AVE dataset is curated for audio-visual event localization and classification. It comprises 4,143 video clips, each lasting 10 seconds, which capture a wide range of common actions and events in daily life. The dataset spans 28 distinct event categories, requiring the model to effectively align audio and visual cues.

**MVSA (Multi-View Sentiment Analysis) (Niu et al., 2016)**   Targeting sentiment analysis in social media content, MVSA combines textual and visual modalities to capture complex emotional expressions. The dataset consists of image-text pairs collected from Twitter. Each pair is annotated with one of three sentiment labels: *Positive, Neutral,* or *Negative*, making it a standard benchmark for image-text sentiment classification.

**IEMOCAP (Busso et al., 2008)**   IEMOCAP is a rich multimodal resource created for emotion recognition and speech processing research. It contains approximately 12 hours of audiovisual data recorded during natural, dyadic interactions (both improvised and scripted) between actors. The dataset includes synchronized video, audio, facial motion capture, and text transcriptions. Following common evaluation protocols, we focus on five primary emotions: *Anger, Excitement, Frustration, Neutrality,* and *Sadness.*

*Table 7.* Dataset statistics for the multimodal benchmarks used in our experiments. Each dataset is split into training, validation, and test sets with an approximate ratio of 8:1:1. We report the exact number of samples in each split.

| Split | CREMA-D | AVE | MVSA | IEMOCAP |
|-------|---------|-----|------|---------|
| Train | 6,698 | 3,311 | 3,552 | 5,426 |
| Val | 0 | 402 | 477 | 683 |
| Test | 744 | 402 | 481 | 676 |
| **Total** | **7,442** | **4,115** | **4,510** | **6,785** |

### C.2. Baselines

**Concatenation**  Concatenation fusion integrates multimodal representations by concatenating modality-specific feature vectors along a designated dimension. A single classifier is applied to the fused representation, whose input dimension equals the sum of all encoder output dimensions, and whose output dimension corresponds to the number of target classes.

**Summation**  Summation fusion aggregates multimodal information by independently classifying each modality and summing their unimodal predictions. Each modality-specific encoder is paired with a classifier whose input and output dimensions correspond to the encoder output and class cardinality, respectively. The summed prediction is used to compute the training loss and update all modality-specific components.

**FiLM (Perez et al., 2018)**  Feature-wise Linear Modulation (FiLM) introduces conditional feature modulation for multimodal fusion. One modality (e.g., audio or text) serves as a conditioning signal to modulate the representations of other modalities. The conditioning network outputs affine parameters $\gamma$ and $\beta$, which are applied to transform modality features before classification.

**Bi-Gated (Kiela et al., 2018)**  Bilinear Gated fusion models cross-modal interactions using bilinear pooling combined with a gating mechanism, explicitly capturing multiplicative correlations between modalities.

**OGM (Peng et al., 2022)**  On-the-fly Gradient Modulation (OGM) is an early representative method for addressing modality dominance at the optimization level. It monitors the generalization discrepancy between modalities during training and dynamically down-scales the gradients of the dominated modality to encourage learning from weaker modalities. While effective in mitigating global modality bias, OGM operates at the batch or modality level and relies on coarse gradient statistics, making it less sensitive to fine-grained, sample-level modality imbalance.

**LFM (Yang et al., 2024)**  Learning via Modality Gap (LFM) introduces a learnable modality gap to explicitly model optimization discrepancies between modalities. By dynamically estimating this gap, LFM adjusts the learning process to prevent any single modality from overwhelming the optimization. However, the modality gap is estimated at a global level and does not explicitly account for per-sample fusion dynamics, limiting its ability to handle localized or instance-specific modality laziness.

**MLA (Zhang et al., 2024)**  Multimodal Learning with Alternation (MLA) mitigates modality imbalance by alternately optimizing unimodal and multimodal objectives, preventing any single modality from dominating the learning process.

**Resample (Wei et al., 2024a)**  Resample alleviates lazy modality behavior by dynamically reweighting and resampling training instances based on modality contribution, emphasizing samples where certain modalities are under-utilized.

**Remix (Ma et al., 2025)**  Remix addresses modality laziness by selectively sampling modality-specific features at the instance level and re-mixing training data to rebalance multimodal learning.

**AMST (Henriques e Silva et al., 2025)**  Alternating Multimodal Skip Training (AMST) addresses multimodal imbalance through a skipping strategy guided by modality-wise convergence rates. Modalities that converge faster are temporarily excluded from training, allowing slower modalities to catch up. Although effective in alleviating optimization imbalance, AMST makes hard modality-level decisions and lacks adaptive, sample-level control, which may lead to suboptimal handling of samples with comparable modality contributions.

**MMTM**(Joze et al., 2020)  MMTM employs a multimodal squeeze-and-excitation mechanism to adaptively recalibrate channel-wise features across modalities.

**CentralNet**(Vielzeuf et al., 2018)  maintains a continuous shared central representation that interacts with unimodal features at each layer via learnable weighted aggregation.

**CAEF**  CAEF, proposed in this work, first refines unimodal representations using self-attention and subsequently performs cross-modal fusion through a co-attention mechanism, aiming to better align modality interactions with downstream task requirements.

### C.3. Experimental setup

*Table 8.* Hyperparameter configurations across different datasets. BS: batch size; $E$: number of training epochs; LR: learning rate; WU: warm-up epochs; Step/Ratio: step-based learning rate decay schedule; WD: weight decay; Dim: unified feature dimension.

| Dataset | BS | $E$ | LR | WU | Step | Ratio | WD | $\gamma$ | $\beta$ | $\alpha$ | $E_p$ | Dim |
|---------|----|-----|----|----|------|-------|----|----|----|----|----|----|
| CREMA-D | 64 | 100 | $1.0\times10^{-3}$ | 0 | 50 | 0.1 | $1.0\times10^{-4}$ | 0.15 | 0.0 | 2 | 4 | 512 |
| AVE | 64 | 100 | $1.0\times10^{-3}$ | 10 | 30 | 0.5 | $5.0\times10^{-4}$ | 0.05 | 0.7 | 2 | 2 | 512 |
| MVSA | 64 | 60 | $1.0\times10^{-5}$ | 0 | 30 | 0.5 | $5.0\times10^{-4}$ | 0.10 | 0.9 | 2 | 2 | 768 |
| IEMOCAP | 64 | 60 | $1.0\times10^{-4}$ | 0 | 20 | 0.1 | $1.0\times10^{-4}$ | 0.15 | 0.7 | 1 | 3 | 768 |

We summarize the hyperparameter configurations for all datasets in Table 8. For a fair comparison, all methods share the same backbone architectures, data splits, optimization algorithms, and training schedules.The unified feature dimension is set to 512 or 768 depending on the backbone architecture to ensure consistent multimodal fusion.Unless otherwise specified, each experiment is conducted with the same random seed to minimize variance caused by random initialization. All experiments were implemented on a single NVIDIA V100 GPUs.

All models are optimized using AdamW, with dataset-specific learning rates and training schedules. A step-based learning rate decay is applied consistently across all benchmarks, with more aggressive decay strategies adopted for MVSA and IEMOCAP to enhance training stability. The hyperparameters $\gamma$, $\beta$, and $\alpha$ respectively control the strength of adaptive-margin masking, laziness balancing weight, and the weighting of unimodal loss terms. All three are tuned on the validation set for each dataset to achieve balanced modality contributions.

## D. Analysis

### D.1. Computational Overhead Analysis

*Table 9.* Comparison of training time per epoch (s) across different datasets.

| Dataset | Concat | OGM | LFM | MLA | Resample | Remix | AMST_FULL | AMRe |
|---------|--------|-----|-----|-----|----------|-------|-----------|------|
| CREMA-D | 21.24 | 22.02 | 21.45 | 20.37 | 35.93 | 25.87 | 34.66 | 21.45 |
| AVE | 17.15 | 17.63 | 17.17 | 16.78 | 22.92 | 21.60 | 30.93 | 17.39 |
| MVSA | 36.64 | 37.12 | 36.81 | 36.28 | 56.32 | 38.52 | 64.65 | 36.70 |
| IEMOCAP | 38.73 | 39.99 | 38.94 | 38.64 | 66.50 | 40.95 | 85.41 | 39.04 |

We analyze the computational overhead introduced by Adaptive-Margin Masking and Restoration from both empirical and theoretical perspectives, including training latency and floating-point operations (FLOPs).

**Empirical Training Latency**  We measure the average training time per epoch (s) on a single NVIDIA V100 GPU. As shown in Table 9, AMRe incurs only a minor overhead compared to Concat:

- **Minor Overhead:** On CREMA-D and MVSA, the additional training time is 0.21s and 0.06s per epoch.

- **Largest Dataset:** On IEMOCAP, which has the highest computational demand, the overhead remains below 0.36s per epoch.

These results suggest that the incorrectness and uncertainty estimation, together with mask generation, introduce negligible latency and do not form a computational bottleneck during training.

**Theoretical FLOPs Analysis**   The efficiency of AMRe primarily stems from operating on compact feature representations rather than raw high-dimensional inputs. The computational complexity can be decomposed as follows:

- **Backbone Cost:** The overall FLOPs are dominated by the backbone encoder (e.g., ResNet-18 or BERT), with complexity on the order of $O(L \cdot N^2 \cdot D)$.

- **AMRe Overhead:** The computation of incorrectness and uncertainty involves simple element-wise operations with complexity $O(C)$, where $C$ denotes the number of classes. The masking operation itself is applied over feature dimensions and incurs $O(D)$ complexity.

Since $O(C) + O(D)$ is orders of magnitude smaller than the backbone cost, the additional FLOPs introduced by AMRe are negligible. Moreover, during the **restoration phase**, which is activated every $E_p$ epochs, the masking module is bypassed entirely, resulting in zero additional FLOPs for those epochs.

**Comparison with Optimization-based Methods**   We further compare AMRe with representative optimization-based approaches, including OGM (Peng et al., 2022) and LFM (Yang et al., 2024). As shown in Table 2, two observations support the efficiency of AMRe:

- **Advantage over Data-level Methods:** Optimization-based approaches consistently exhibit substantially lower latency than data-level balancing methods (e.g., Resample, Remix), as the latter incur significant I/O and preprocessing overhead.

- **Efficiency among Optimization Methods:** AMRe achieves comparable or better efficiency relative to OGM and LFM.

    - OGM requires computing gradient norms during backpropagation, introducing additional matrix operations.
    - LFM relies on estimating learnable modality gaps or maintaining global statistics.
    - AMRe uses lightweight metric computation ($O(C)$) and simple feature masking ($O(D)$). The periodic restoration mechanism further reduces overhead by disabling masking for a substantial portion of training.

### D.2. Hyperparameters

To assess the robustness of AMRe and the sensitivity of its design choices, we analyze three key hyperparameters: the balancing weight $\beta$, the soft margin threshold $\gamma$, and the restoration interval $E_p$. For completeness, all results are reported in this appendix.

*Table 10.* Sensitivity analysis of $\beta$ and $\gamma$ across four datasets. **Bold** indicates the best results.

| Dataset | $\beta$ Value | | | | | | | $\gamma$ Value | | | | | |
| --- | --- | --- | --- | --- | --- | --- | --- | --- | --- | --- | --- | --- | --- |
| | **0** | **0.1** | **0.3** | **0.5** | **0.7** | **0.9** | **1.0** | **0** | **0.05** | **0.10** | **0.15** | **0.20** | **0.25** |
| **CREMA-D** | **80.78** | 76.88 | 78.36 | 76.21 | 76.74 | 74.60 | 75.94 | 79.84 | 78.90 | 78.09 | **80.78** | 77.55 | 78.94 |
| **AVE** | 73.63 | 72.89 | 71.89 | 75.62 | **76.37** | 75.37 | 75.12 | 75.37 | **76.37** | 75.87 | 75.87 | 75.83 | 75.37 |
| **MVSA** | 73.60 | 73.18 | 73.18 | 74.01 | 75.47 | **75.68** | 74.01 | 74.43 | 74.22 | **75.68** | 73.60 | 74.64 | 74.84 |
| **IEMOCAP** | 67.31 | 66.12 | 67.46 | 66.42 | **68.93** | 65.53 | 65.83 | 66.42 | 67.46 | 65.38 | **68.93** | 65.68 | 65.38 |

**Laziness Balancing Weight**   Table 10 shows the trade-off between incorrectness ($\beta \to 1$) and uncertainty ($\beta \to 0$) in identifying lazy modalities. The results reveal a distinct divergence rooted in task characteristics:

- **Uncertainty-Dominant Task (CREMA-D)** This audio-visual emotion dataset lacks explicit semantic cues and benefits most from $\beta = 0$, where uncertainty serves as a more reliable signal under noisy predictions.

- **Incorrectness-Dominant Tasks (IEMOCAP, AVE, MVSA)** Tasks with fast-learning or strongly informative modalities (e.g., text or salient visual objects) prefer larger $\beta$ values (0.7–0.9). In these settings, loss discrepancy offers a clearer indication of imbalance.

The choice of $\beta$ is dataset-dependent and plays a critical role in effectively identifying and mitigating modality laziness.

**Soft Margin**    Table 10 shows that introducing a non-zero soft margin generally improves performance over hard masking ($\gamma = 0$). The preferred margin varies across datasets and appears related to data ambiguity:

- **Higher Ambiguity:** Datasets with subjective or noisy labels (e.g., CREMA-D and IEMOCAP) favor larger margins (around $\gamma = 0.15$), which help reduce unstable masking decisions near the boundary.

- **Lower Ambiguity:** Datasets with clearer semantic cues (e.g., AVE and MVSA) perform best with smaller margins ($\gamma \in [0.05, 0.10]$), where masking remains more selective.

*Table 11.* Impact of the restoration interval $E_p$ (%). **Bold** denotes the best results.

| $E_p$ | CREMA-D | AVE | MVSA | IEMOCAP |
|---|---|---|---|---|
| 0 | 73.52 | 72.89 | 73.18 | 65.38 |
| 1 | 77.55 | 71.80 | 73.39 | 67.90 |
| 2 | 80.65 | **76.37** | **75.68** | 65.24 |
| 3 | 80.65 | 73.13 | 73.60 | **68.93** |
| 4 | **80.78** | 75.62 | 74.22 | 65.58 |
| 5 | 79.30 | 73.88 | 73.80 | 67.16 |

**Restoration**    Table 11 examines the effect of the restoration interval by comparing full optimization ($E_p = 0$), AMRe w/o restoration ($E_p = 1$), and periodic restoration ($E_p \geq 2$).

- **Inverted U-Shape Trend.** Performance follows a clear inverted U-shaped pattern as $E_p$ increases. AMRe w/o restoration ($E_p = 1$) already improves upon the baseline by alleviating modality dominance, but extended freezing of the dominated modality eventually leads to representational degradation. Periodic restoration mitigates this effect by reintroducing full joint optimization at regular intervals, with the strongest results consistently observed for moderate intervals ($E_p \in [2, 4]$). When the interval becomes too large, performance degrades again, supporting the view that overly prolonged suppression disrupts feature balance and harms multimodal fusion.

- **Task Dependence.** The optimal interval depends on task characteristics. Ambiguity dominated datasets such as CREMA-D benefit from longer masking phases (typically $E_p = 4$), which encourage the model to exploit more subtle cues. In contrast, datasets with clearer semantic structure, such as AVE, require more frequent restoration (typically $E_p = 2$) to maintain stable and informative representations.

