# OpenReview forum: "Adaptive-Margin Masking and Restoration for Balanced Multimodal Learning"
_ICML.cc/2026/Conference — Submitted to ICML 2026_

### Official Review · Reviewer_TZL5 · 2026-02-24

**Soundness:** 3
**Presentation:** 3
**Significance:** 2
**Originality:** 3
**Overall Recommendation:** 4
**Confidence:** 4

**Summary:**

This paper focuses on multimodal imbalanced learning, a critical research area. Addressing two limitations in the field—1) The identification of lazy modality is incomplete and overly sharp; 2) The dominated modality can be lagged by under-optimized lazy modalities—the authors propose AMRe. This method designs a novel dominant mode evaluation strategy and enhances the model's attention to underrepresented modalities through two mechanisms: cyclically masking dominant modes and fully learning all modalities. It integrates alternating training with optimization-based strategies. Experiments demonstrate that AMRe achieves state-of-the-art performance across multiple datasets.

**Compliance With Llm Reviewing Policy:**

Affirmed.

**Final Justification:**

The author solved my all concerns.

**Key Questions For Authors:**

1. The first contribution of this paper is adaptive-margin modality identification, where the authors claim to leverage incorrectness and uncertainty to evaluate modality inertia. However, incorrectness is measured using cross-entropy between a single-modality classification head and labels, while uncertainty introduces KL divergence. These two independent evaluation strategies have been thoroughly demonstrated and utilized in prior work. Adaptive-margin modality identification merely combines these two metrics using a single weighting factor, significantly diminishing the paper's contribution. The authors are advised to further explain how adaptive-margin modality identification differs from existing work beyond integrating two established metrics. This clarification would facilitate a reassessment of the research contribution's value.

2. The authors employ a masking strategy to periodically obscure the dominant modality, thereby enhancing the model's attention to the underrepresented modality. Mask strategies have been extensively studied in related fields (multimodal imbalanced learning, modality dropout) (e.g., sample-level masking commonly used in modality dropout tasks, feature-space masking in multimodal imbalanced learning (for example, OPM)). I recommend supplementing experiments to further demonstrate AMRe's performance advantages over existing strategies.

3. Table 4 presents parameter sensitivity analysis. However, we observe that different datasets exhibit sensitivity to these hyperparameters, particularly $\beta$. On the CREMAD dataset, the authors even set $\beta=0$ to achieve optimal results. This implies that the incorrect branch in Equation 8 is entirely blocked. Under such circumstances, it may give readers the impression that the experimental improvements depend heavily on empirical parameter tuning rather than the method itself. Therefore, we hope the authors provide analysis and clarification on this issue.

4. I recommend the authors add a Limitations section to evaluate the method's constraints and provide directions for future research.

Overall, the paper's focus on addressing mode imbalance is commendable, and its integration of gradient-based methods with alternating optimization strategies represents an advanced research perspective. However, I retain reservations regarding the contribution statement and experimental sections, leading me to express a conservative opinion. I encourage the authors to address these concerns in their rebuttal.

**Limitations:**

See questions.

**Strengths And Weaknesses:**

**Strengths:**

1. The research content of this paper is advanced, and multimodal imbalance represents a critical challenge for multimodal models. Multimodal imbalanced learning methods hold promise for further enhancing the performance of multimodal models.
2. The paper is written fluently, adhering to standard writing conventions. Its illustrations are visually appealing and accessible to a broad audience.
3. The experiments are comprehensive, providing substantial experimental evidence that diversely supports the feasibility of the proposed AMRe approach.

**Weakness:**

See questions.

---

> ### Author Rebuttal · Authors · 2026-03-31
>
> **Weakness 1:** lacks of contribution's values.
>
> **Response 1:**
>
> + **Dual-dimension evaluation**: We agree that Cross-Entropy (CE) and KL divergence are classic metrics. However, AMRe's innovation lies in conceptualizing modality laziness from two distinct, complementary dimensions: Incorrectness (result-oriented) and Uncertainty (confidence-oriented). CE and KL are merely instantiations of this concept. To prove that our gains stem from this dual-dimension philosophy rather than the specific metrics, we have supplemented experiments using alternative metrics(e.g., replacing CE with Brier Score), **as shown in Table 5**. The consistently superior results confirm that it is the dual-perspective framework itself that provides a more robust measurement space, rather than a coincidental combination of CE and KL.
> + **Soft identification**: AMRe reveals the **overly-sharp** lazy identification problem in existing works. Prior methods force a rigid binary classification even when modalities are actually in a quasi-balanced state (e.g., with marginal laziness score differences, such as 0.51 vs. 0.49), the rigid decision may incorrectly degrade a balanced modality into a lazy one. AMRe introduces a soft margin $\gamma$ specifically to identify these **ambiguous samples** and grant them immunity from masking, because treating ambiguous samples as imbalanced introduces noise into the optimization (Line 069, left). Please refer to our response 6 to Reviewer n9zw for the **instance analysis**.
>
> **Table 5**.  Result(\%) of different laziness evaluation metrics. Specifically, CE Loss and Brier Score are utilized to measure incorrectness, whereas KL divergence and Entropy are employed to evaluate uncertainty.
>
> |     Method      |    AVE    |   MVSA    |  IEMOCAP  |
> | :-------------: | :-------: | :-------: | :-------: |
> |     CE loss     |   75.12   |   74.01   |   65.83   |
> |      Brier      |   75.87   |   74.84   |   66.42   |
> |       KL        |   72.39   |   73.60   |   66.72   |
> |     Entropy     |   74.88   |   71.93   |   66.12   |
> |   CE loss+KL    | **76.37** | **75.68** | **68.93** |
> | CE loss+Entropy |   74.88   |   74.22   |   67.75   |
> |    Brier+KL     |   74.13   |   74.22   |   66.72   |
> |  Brier+Entropy  |   73.88   |   74.84   |   65.38   |
> |                 |           |           |           |
> > 1) **CE loss:** CE loss measures the logarithmic discrepancy between predictions and true labels;
> 2) **Brier Score:** As a strictly proper scoring rule, the Brier Score measures the mean squared error between predicted probabilities one-hot targets;
> 3) **Information Entropy:** Entropy quantifies the uncertainty or dispersion of the predicted distribution;
> 4) **Kullback-Leibler (KL) Divergence:** KL divergence measures the relative distance between the predicted distribution and uniform distribution
>
> &nbsp;
>
> **Weakness 2:** comparison to previous existing works.
>
> **Response 2:**
>
> Please refer to the **responses to Weakness 1 for Reviewer rMSX**.
>
> &nbsp;
>
> **Weakness 3:** hyperparameter sensitivity.
>
> **Response 3:**
>
> We acknowledge that the performance is sensitive to $\beta$. However, this sensitivity fundamentally stems from the varying intrinsic level of incorrectness and uncertainty inherent in different datasets, which prior works have neglected. The proposed dual-dimensional measurement is precisely designed to enable our method to adapt to various datasets, and we have **analyzed this dataset-dependent behavior in detail in Appendix D.2**(lines 874-882). While the current work uses a simple weighted combination, we plan to explore fully adaptive fusion methods for these two dimensions in our future work.
>
> &nbsp;
>
> **Weakness 4:** limitation section
>
> **Response 4:**
>
> We will include an assessment of the limitations in the next version.
>
> + **Limitations:** Although the proposed AMRe framework effectively overcomes the rigid and overly-sharp of existing laziness identification methods, and circumvents the optimization dominance of dominated modalities via sample-level precise masking, there remains room for future exploration. First, the current joint evaluation metric relies on a static fusion paradigm. A promising future direction is to introduce a dynamic, data-driven weighting mechanism to **adaptively fuse** the two evaluation dimensions during training. Second, while our epoch-level periodic restoration strategy has proven empirically effective, its granularity can be further refined. Future work could investigate **gradient-aware or other fine-grained adaptive** mechanisms to maximize the efficacy of the restoration phase.

---

> > ### Author Rebuttal · Reviewer_TZL5 · 2026-04-01
> >
> > Thank you to all the authors for their careful responses, and I'm also glad that they could agree with my suggestions regarding the manuscript. The authors addressed my concerns, so I decided to change my rating to 4.

---

> > > ### Author Response · Authors · 2026-04-04
> > >
> > > Thank you very much for your reply. Your suggestions have provided us with great help in our work.

---

### Official Review · Reviewer_n9zw · 2026-02-24

**Soundness:** 3
**Presentation:** 3
**Significance:** 3
**Originality:** 2
**Overall Recommendation:** 4
**Confidence:** 4

**Summary:**

This work studies the problem of modality laziness in multimodal learning and proposes Adaptive-Margin Masking and Restoration (AMRe) to balance dominated and lazy modalities during training. The method combines a laziness score based on incorrectness and uncertainty with a soft margin and a periodic restoration phase.

**Compliance With Llm Reviewing Policy:**

Affirmed.

**Final Justification:**

After considering both the paper and the authors’ rebuttal, I find that the work is generally sound and clearly presented, with reasonable significance. The rebuttal effectively addressed most of my technical concerns, particularly regarding the epoch-level restoration strategy and the distinction between under-optimized and noisy samples, which led me to raise my score. However, my reservations about the originality of the approach remain largely unresolved and continue to weigh against a higher rating

**Key Questions For Authors:**

A modality cannot always be defined as lazy or dominated by another one. Under perfect labeling and perfect modality–class alignment, the assumptions made in this work are valid. However, in cases where one modality provides limited information or when datasets are not well curated, the situation becomes more complex. For example, if the audio signal of a sample is very noisy, the audio modality could be flagged as lazy. Forcing the audio encoder to extract useful information from such a noisy sample may be ineffective and potentially harmful.

How does the proposed laziness identification mechanism distinguish between a truly under-optimized (lazy) modality and a modality that is inherently noisy or uninformative for a given sample/task?

Is the restoration applied at the epoch level? If E_p=2, does this mean one full epoch with masking followed by one full epoch without masking?
If so, why is restoration designed at the epoch level? Have the authors considered applying restoration within each epoch, for example at the batch level?

**Strengths And Weaknesses:**

Soundness:

* The notation throughout the paper should be revised and improved. It is currently inconsistent. In Section 3.1, the framework is introduced with a generic number of modalities M, but the theoretical derivation later focuses only on two modalities (rows 143–154, left column). The formulation then switches back to M modalities again (rows 117–122, right column). The paper would benefit from a single, coherent formulation for the general M-modality case, with the two-modality scenario presented only as a special case if necessary.

* The theoretical derivation in Section 3.2, which is central to the paper, is not clearly explained. Some components are not properly characterized in the main text. For example, C is used without explicitly reminding the reader that it denotes the number of classes, and the definition of pi,j is not sufficiently clear unless the reader carefully checks the appendix. Important quantities, such as the softmax probabilities, should be clearly defined in the main paper. As written, the derivation is difficult to follow and not fully convincing.

* A modality cannot always be defined as lazy or dominated by another one. Under perfect labeling and perfect modality–class alignment, the assumptions made in this work are valid. However, in cases where one modality provides limited information or when datasets are not well curated, the situation becomes more complex. For example, if the audio signal of a sample is very noisy, the audio modality could be flagged as lazy. Forcing the audio encoder to extract useful information from such a noisy sample may be ineffective and potentially harmful.

Presentation:

* Overall, the narrative is easy to follow. However, there are issues with the notation (see the Soundness section).

* Figure 1 is not easy to understand at first glance. It is difficult to grasp the method by only looking at the figure. The caption is not sufficiently descriptive and does not fully clarify the visualization.

* Equation 3 and Equation 12 use the same symbols while referring to slightly different training settings. This creates confusion and gives the impression that the meaning of symbols changes during the paper. It would improve readability to introduce distinct notation for the mask-conditioned loss.

Significance:
* The paper addresses a relevant and practical problem in multimodal learning. The proposed method is simple, lightweight, and can be integrated into existing fusion pipelines, which increases its practical utility.

* The impact is likely moderate. While the empirical improvements are consistent, the conceptual advance over recent imbalance-aware methods is incremental.  Within the multimodal representation learning area this approach could be useful and inspire further refinements of adaptive masking strategies.

Originality:

The framework proposed in this paper builds upon recent trends in multimodal learning. In particular, it extends the work of "Wei et al. (2024a)" and "Ma et al. (2025)" by integrating their ideas and introducing a new method to mask dominating modalities. It also alternates masking with standard joint learning to prevent representation degradation caused by excessive masking.

---

> ### Author Rebuttal · Authors · 2026-03-31
>
> **Weakness 1:** the notation throughout the paper should be revised and improved.
>
> **Response 1:**
>
> We will improve the derivation to M modalities in the next version
>
> &nbsp;
>
> **Weakness 2**: theoretical derivation need clearly explanation.
>
> **Response 2:**
>
> We will improve the derivation and move them from the appendix to the main text.
>
> &nbsp;
>
> **Weakness 3**: modality that is flagged as lazy because of noise can harm the model.
>
> **Response 3:**
>
> Please refer to the response of **weakness 6**.
>
>
>
> &nbsp;
>
>
>
> **Weakness 4:** Figure 1 is not sufficiently descriptive and visualized.
>
> **Response 4:**
>
> We will refactor Figure 1 and its caption in the revised version to enhance its self explanatory power. The updated Figure 1 is available at https://anonymous.4open.science/r/Rebuttal-3D2C.
>
>
>
> &nbsp;
>
>
>
> **Weakness 5:** Equation 3 and 12 are misleading.
>
> **Response 5:**
>
> To resolve the notation ambiguity, we have explicitly distinguished the loss functions for the two training settings.
> + **The standard full optimization loss** (Eq. 3) is now formally denoted as $L_{std} = L^f + \alpha \cdot \sum_{m_j=1}^M L^{m_j}$, where $L^{m_j}$ is the traditional unimodal loss (Eq. 2).
> + **The masking loss** (Eq. 12) is updated to $L_{AMRe} = L^f + \alpha \cdot \sum_{m_j=1}^M \widetilde{L}^{m_j}$, utilizing $\widetilde{L}^{m_j}$ to clearly represent the mask-conditioned unimodal loss (Eq. 11).
>
> &nbsp;
>
>
>
> **Weakness 6**: difference between under-optimized (lazy) modality samples and noisy samples.
>
> **Response 6:**
>
> To demonstrate how AMRe avoids learning from noise, we analyze representative samples from our AVE training (see provided code link, left ate line 33). As shown in **Table 3**, compared to traditional multimodal masking identification, AMRe shows distinct advantages:
>
> + **Ambiguous Samples**:
>
>   In the "Ambiguous", both modalities exhibit hight and similar probabilities. Traditional methods rigidly mask $m_2$, AMRe's soft margin ($\gamma$) detects the negligible gap and **applies no mask**, preventing unnecessary optimization.
>
> + **Noise Samples**:
>
>   (1) **In Noise 1**, incorrectness-based masking misidentifies $m_1$ as the dominant modality, whereas uncertainty-based masking targets $m_2$. AMRe's 2D metric identifies $m_2$ as a noisy modality (high error, low uncertainty) and masks it to sever its harmful gradients.
>
>   (2) **In Noise 2**, traditional methods blindly mask the dominant $m_2$ to ``force'' $m_1$ to learn. Recognizing $m_1$'s severe lack of information, AMRe instead masks $m_1$ to isolate the noise, allowing the healthy modality to guide the learning.
>
> **Table 3.** Analysis of masking decisions on AVE Dataset different sample types . “Inc” and “Unc” denote traditional single-metric masking based on incorrectness and uncertainty, respectively. $m_j$ is masking modality $j$, $None$ is not masking modality. $h^{m_1}_i$ is calculation Eq. 8.
>
> | Sample Type | $m_1$ Prob. | $m_2$ Prob. | $h^{m_1}_i$ | $h^{m_2}_i$ | AMre   | Inc   | Unc   |
> | ----------- | ----------- | ----------- | :---------: | :---------: | ------ | ----- | ----- |
> | Ambiguous   | 0.8816      | 0.8677      |   0.45505   |   0.54495   | $None$ | $m_2$ | $m_2$ |
> | Noise 1     | 0.2152      | 0.0355      |   0.70353   |   0.30034   | $m_2$  | $m_1$ | $m_2$ |
> | Noise 2     | 0.0938      | 0.6330      |   0.20263   |   0.79737   | $m_1$  | $m_2$ | $m_2$ |
>
>
>
>
>
>
>
> &nbsp;
>
>
>
> **Weakness 7:** why not batch-level.
>
> **Response 7:**
>
> The epoch-level restoration strategy proposed in our paper is primarily motivated by representation stability. Our goal is to allow the model a full epoch to recalibrate and align its optimization direction across a diverse combination of samples (as supported by Figure 3(a)).
>
> Following your suggestion, we also experimented with batch-level restoration, as shown in **Table 4**. However, we found that such **high-frequency switching** leads to degraded overall performance.
>
> **Table 4**. Result(\%) of different restoration strategies. Epoch-level is AMRe.
>
> |     Method      |  CREMAD   |    AVE    |   MVSA    |  IEMOCAP  |
> | :-------------: | :-------: | :-------: | :-------: | :-------: |
> |     Concat      |   73.52   |   72.89   |   73.18   |   65.38   |
> | w/o Restoration |   78.76   |   75.37   |   74.43   |   66.42   |
> |   Batch-level   |   73.66   |   73.88   |   74.22   |   65.83   |
> |   Epoch-level   | **80.78** | **76.37** | **75.68** | **68.93** |

---

> > ### Author Rebuttal · Reviewer_n9zw · 2026-04-03
> >
> > Thank you to the authors for taking the time to address my concerns. Most of the issues have been satisfactorily resolved, particularly those related to the epoch-level restoration strategy and the distinction between under-optimized (lazy) and noisy samples.
> >
> > I will increase my score to 4. However, I am not inclined to rate it higher, as I still have some reservations regarding the originality of the work, which remain a significant concern.

---

> > > ### Author Response · Authors · 2026-04-04
> > >
> > > Response 1: Thank you very much for your reply and score increase, the one key contribution is that AMRe is designed **without losing generality on the numbers of input modalities** theoretically and empirically.
> > >
> > > &nbsp;
> > >
> > > The generalized derivation for $M$ modalities is formally defined as follows:
> > >
> > > **1. General Multimodal Framework:**
> > >
> > > Let the dataset be defined as $\mathcal{D} = \{(x_i, y_i)\}_{i=1}^N$, where $N$ is the number of samples. Each sample $x_i = (x_i^1, x_i^2, \dots, x_i^M)$ consists of $M$ modalities, and $y_i \in \\{1, 2, \dots, C\\}$ is the ground-truth label for $C$ categories.
> > >
> > > For modality $m\in(1,2,\dots,M)$, its input is processed by the corresponding encoder $z^m_i=\varphi^m(x^m_i;\theta^m)\in \mathbb{R}^{d_{\varphi_m}}$, where $\theta^m$ represents the parameters of encoder $\varphi^m$, and $d_{\varphi^m}$ is output dimension of encoder $\varphi^m$.
> > >
> > > The multimodal features are concatenated to obtain a joint representation $z^f_i=[z^1_i,z^2_i,\dots,z^m_i]$ .With the above notions, the logit output of the multimodal model can be formulated as :
> > > $$
> > > f(x_i) = \psi_f(z^f)=W\cdot{z^f_i}+b=W^1 \cdot z^1_i + W^2 \cdot z^2_i+\dots+ W^M \cdot z^M_i+b \tag{re.1}
> > > $$
> > > where $f(x_i) \in \mathbb{R}^C$ and the classifier weight matrix is partitioned as $W = [W^1,W^2,\dots,W^M],W^m\in \mathbb{R}^{C \times d_{\varphi^m}}$.The standard cross-entropy loss for sample $i$ is
> > >
> > > $$
> > > \mathcal{L}^f(x_i, y_i) = - \log \left( \frac{e^{f(x_i)^{y_i}}}{\sum_{r=1}^C e^{f(x_i)^r}} \right) \tag{re.2}
> > > $$
> > >
> > > &nbsp;
> > >
> > > **2. Gradient Interference Analysis:**
> > >
> > > The gradient of the loss with respect to the feature $z^m_i$ of modality $m$ is derived via the chain rule :
> > > $$
> > > \frac{\partial \mathcal{L}^f(x_i,y_i)}{\partial z^m_i} =  \frac{\partial \mathcal{L}^f(x_i,y_i)}{\partial z^f_i} \cdot \frac{\partial z^f_i}{\partial z^m_i}
> > > =\sum^C_{j=1}(softmax(f(x_i)^j)-\mathbb I(j=y_i)) \cdot W^m_j \tag{re.3}
> > > $$
> > > To explicitly reveal the gradient interference from the dominated modality, we expand the softmax term $\text{softmax}(f(x_i)^j)$ as follows:
> > >
> > > $$
> > > softmax(f(x_i)^j) = \frac{e^{f(x_i)^j}}{\sum_{r=1}^C e^{f(x_i)^r}}=\frac{e^{(W^1_j\cdot z^1_i+\dots+W^m_j\cdot z^m_i+\dots+W^M_j\cdot z^M_i+b_j)}}
> > > {\sum_{r=1}^C e^{(W^1_r\cdot z^1_i+\dots+W^m_r\cdot z^m_i+\dots+W^M_r\cdot z^M_i+b_r)}}
> > > \\\\=\frac{e^{(W^m_j \cdot z^m_i + b_j)}P^m_{i,j}}
> > > {\sum_{r=1}^C (e^{(W^m_r \cdot z^m_i + b_r)}P^m_{i,r})} \tag{re.4}
> > > $$
> > >
> > > where $P^m_{i,j} = \prod_{k \neq m}^M \rho^k_{i,j}$ represents the **interference term** from all other modalities, and $\rho^k_{i,j} = e^{W^k_j \cdot z^k_i}$ is defined as the **interaction factor** introduced by modality $k$.
> > >
> > > &nbsp;
> > >
> > > **3. AMRe Maksing & Degeneration**
> > >
> > > When AMRe identifies a dominated modality $m^\*$ (via Eq. 10) and masks its features to zero ( $z^{m^\*}_i = 0$ ), its corresponding interaction factor mathematically strictly evaluates to
> > >
> > > $$
> > > \rho^{m^*}_{i,j} = e^{W^{m^\*}_j \cdot 0} = 1. \tag{re.5}
> > > $$
> > >
> > >
> > > This active sample-level masking operation **physically cuts off the gradient magnitude** passed from the dominant modality $m^\*$ to the lazy modality $m$.
> > >
> > > Consequently, the updated interference term becomes $\tilde{P}^m_{i,j} = \prod_{k \notin \\{m, m^*\\}}^M \rho^k_{i,j}$, and Eq.(re.4) for optimizing modality $m$ updates to:
> > >
> > > $$
> > > \text{softmax}(f(x_i)^j) = \frac{e^{W^m_j \cdot z^m_i + b_j} \tilde{P}^m_{i,j}}{\sum_{r=1}^C e^{W^m_r \cdot z^m_i + b_r} \tilde{P}^m_{i,r}} \tag{re.6}
> > > $$
> > >
> > > Particularly, in a standard dual-modality scenario ($M=2$), masking the dominated modality results in $\tilde{P}^m_{i,j} = \mathbf{1}$. Under this condition, the softmax term (Eq.(re.6)) gracefully degenerates into a pure unimodal computation. This entirely eliminates the dominated modality's gradient interference, allowing the lazy modalities to optimize efficiently.

---

### Official Review · Reviewer_rMsX · 2026-03-09

**Soundness:** 3
**Presentation:** 2
**Significance:** 3
**Originality:** 2
**Overall Recommendation:** 3
**Confidence:** 3

**Summary:**

This paper proposes Adaptive-Margin Masking and Restoration (AMRe), a method designed to address the problem of modality laziness in multimodal learning. AMRe aims to achieve balanced multimodal training, enhancing the robustness and efficiency of models in multimodal tasks.

**Compliance With Llm Reviewing Policy:**

Affirmed.

**Key Questions For Authors:**

(1)The rationale for using incorrectness and uncertainty to distinguish between lazy and dominant modalities requires further analysis. Have the authors considered cases where a modality exhibits a low error rate but high uncertainty, or a high error rate but low uncertainty? How would AMRe handle such scenarios?

(2)In the paper, it is stated that “the dominated modality is masked to promote optimization of the lazy one.” However, in Figure 1(b), masking operations appear to be applied to both modalities. The authors should clarify this apparent discrepancy.

(3)In the Related Work section, existing alternating optimization works on “masking multi-modal inputs” are mentioned. How does the proposed Training Strategy differ from, or improve upon, these prior masking-based approaches? Please provide a more detailed and essential explanation.

(4)Notation Issues:
(i) Equations (3) and (12) are formally identical, but based on the context they are not exactly the same. It is recommended to use distinct symbols to differentiate them.
(ii) In Algorithm 1, the input fusion head ψ_m is denoted differently from ψ_f in the main text. The notation should be made consistent.

**Limitations:**

The proposed method relies on several key hyperparameters whose optimal values vary significantly across different datasets. When applied to new tasks or datasets, substantial hyperparameter tuning may be required, which could reduce the method’s usability and ease of deployment.

**Strengths And Weaknesses:**

Strengths:
1. The paper is well-structured, with a reasonable method design and thorough experimental setup, demonstrating broad applicability across multimodal tasks.

2. The proposed approach integrates incorrectness and uncertainty to identify lazy modalities, and employs an adaptive-margin modality identification with threshold γ to softly identify dominated and lazy modalities. This helps alleviate issues caused by ambiguous samples near decision boundaries that may negatively affect multimodal training.

- Weaknesses:
1. The distinction or improvement of the proposed training strategy compared to existing “masking multi-modal inputs” approaches is not clearly articulated.

2. The paper lacks analysis on the convergence or gradient stability of the masking optimization and restoration mechanisms.

---

> ### Author Rebuttal · Authors · 2026-03-30
>
> **Weakness 1:** comparison to existing ``masking multi-modal inputs" approaches.
>
>  **Response 1**:
>
> Although masking strategies have been widely studied, the core contribution of AMRe lies in **sample-level dominated modality masking.** AMRe further provides a systematic solution to the overlooked problem of feature degradation.
>
> + **Dominated modality masking**: Existing methods like Remix and MLA try to solve the issue by splitting the dataset into single-modality subsets and training them separately. However, this causes the model to lose the original multimodal fusion features. **AMRe does not modify the input data.** Instead, it applies precise, sample-level masking during training based on mathematical derivation. This approach effectively **cuts off dominated modality gradient** interference while fully preserving the model's ability to learn multimodal fusion.
> + **Deterministic Intervention**: Existing sample-level masking methods (like Random Dropout and OPM) rely on **fixed or dynamic probabilities**, which essentially act like random guessing. This probabilistic dropping can easily hurt the already lazy modality by reducing its update signals (As shown in the **Table 2** that our supplemented experiments, where their performance only slightly improved or even dropped). Instead of "rolling the dice" like OPM [1], **AMRe introduces a deterministic, adaptive identification** mechanism based on incorrectness and uncertainty. We also introduce a soft margin ($\gamma$) to actively protect "ambiguous samples" from unnecessary masking.
> + **Feature Degradation**: Prior work ignore that suppressing the dominated modality for a long time causes it to lack sufficient optimization, it may degrades into a lazy one.AMRe formally reveals this feature degradation issue. By introducing a **epoch-level periodic restoration** mechanism that alternates between masking and full optimization, AMRe mitigates lazy modality laziness without imposing excessive pressure the dominated modality. The detailed different restoration method analysis are provided in our **response 7 to Reviewer n9zw**.
>
> **Table 2**. Results(\%) of the of modality dropout methods.
>
> | Method |  CREMA-D  |    AVE    |   MVSA    |  IEMOCAP  |
> | :----: | :-------: | :-------: | :-------: | :-------: |
> | Concat |   73.52   |   72.98   |   73.18   |   65.38   |
> | Random |   74.33   |   72.14   |   73.18   |   65.53   |
> |  OPM   |   78.36   |   74.38   |   72.98   |     -     |
> |  AMRe  | **80.78** | **76.37** | **75.68** | **68.93** |
>
> >  [1] Wei, Y., Hu, D., Du, H., & Wen, J. R. (2024). On-the-fly modulation for balanced multimodal learning. *IEEE Transactions on Pattern Analysis and Machine Intelligence*, *47*(1), 469-485.
>
> &nbsp;
>
> **Weakness 2**: convergence or gradient stability analysis
>
> **Response 2:** We provide both theoretical and empirical evidence for AMRe's stability and convergence. Theoretically, we prove via the chain rule that masking decouples the interaction term to eliminate gradient interference (lines 158-162), which is empirically supported by the high gradient cosine similarity in Fig 3(a). Furthermore, while the alternating masking and restoration phases introduce local fluctuations in the training curve (Fig 3(c)), these "jumps" effectively lead to a better convergence point. Finally, our ablation studies (Tables 3 \& 11) demonstrate that periodic restoration ($E_p \ge 2$) prevents irreversible feature degradation caused by prolonged suppression (lines 430-432), serving as the essential cornerstone of overall training stability.
>
> &nbsp;
>
> **Weakness 3:** reasons of using incorrectness and uncertainty for laziness identification:
>
> **Response 3:**
>
> Due to space limitations, please refer to our Response 6 to Reviewer n9zw for the **instance analysis**, and to our **Response 1 to Reviewer TZL5** (Dual-dimension evaluation) for the **parameter analysis**.
>
> &nbsp;
>
> **Weakness 4:** misunderstanding of masking operations
>
> **Response 4:**
>
> In the right panel of Fig. 1(b), each column corresponds to a different sample, and the white regions denote masked features. For better visual clarity, we use different geometric shapes to distinguish the features of different samples. The updated Fig. 1(b) is available at https://anonymous.4open.science/r/Rebuttal-3D2C.
>
> &nbsp;
>
> **Weakness 5:** notation Issues:
>
> **Response 5:**
>
> We will rectify them in the next version. We have explicitly distinguished the loss functions:
>
> + **Equation 3** has been updated to $L_{std} = L^f + \alpha \cdot \sum_{m_j=1}^M L^{m_j}$, where $L^{m_j}$ is the traditional unimodal loss (Eq. 2).
>
> + **Equation 12** has been updated to $L_{AMRe} = L^f + \alpha \cdot \sum_{m_j=1}^M \widetilde{L}^{m_j}$, where  $\widetilde{L}^{m_j}$ clearly represent the mask-conditioned unimodal loss (Eq. 11).

---

> > ### Author Rebuttal · Reviewer_rMsX · 2026-04-06
> >
> > Thank you for the author's reply. Overall, I think a score of 3 is appropriate for this manuscript. So I decided to keep the original score.

---

> > > ### Author Response · Authors · 2026-04-06
> > >
> > > I'm glad to have solved all your problems. We appreciate if you can explicitly comment out the concerns that make you keep the score.

---

### Official Review · Reviewer_PKBW · 2026-03-12

**Soundness:** 2
**Presentation:** 3
**Significance:** 2
**Originality:** 2
**Overall Recommendation:** 3
**Confidence:** 5

**Summary:**

This paper studied how to avoid specific modalities from dominating the training process. The key idea is first to identify which modalities are dominating, then mask out those modalities for the next round of training. Modality masking and full optimization are iteratively applied during training. Experiments are conducted on three multimodal benchmarks, covering modalities like image, text, and audio.

**Compliance With Llm Reviewing Policy:**

Affirmed.

**Final Justification:**

While the authors provide a theoretical framework and useful unimodal benchmarks, the performance improvements remain marginal and do not yet clearly justify the additional complexity compared with the baselines. In addition, the method’s reliance on explicit unimodal signals raises concerns about its transferability to current large-scale VLM architectures.

**Key Questions For Authors:**

Please refer to the weakness.

**Limitations:**

Discussion on scalability should be included.

**Strengths And Weaknesses:**

**Steongths**
+ The preliminary section clearly describes the problem and provides sufficient information for model laziness.
+ Multiple aspects of the proposed method have been ablated.

**Weaknesses**
+ The proposed method is heuristic and lacks grounding in established theory. And there is no new concept/insight/problem introduced, which makes the paper read incrementally.
+ Some results (e.g., Table 2) are marginally improved compared to the baseline methods.
+ The scalability issue is not discussed. Can the proposed method effectively handle high-modality scenarios (> 5 modalities)?
+ It would be better to report the unimodal performance for each dataset. The goal to tackle modalities' lazeness is to fully utilize all modalities. Reporting the unmodal performance can make it clear to show the advantage of the proposed method.
+ The studied backbones are limited to “small” models (compared to current VLMs). Is modality laziness still an issue for current billion-level VLMs? Any empirical evidence from the author or existing literature?
+ Can random modality masking solve the modality laziness issue? What’s the performance compared to the proposed method?

---

> ### Author Rebuttal · Authors · 2026-03-30
>
> **Weakness 1:** lacks grounding in theory and no new concept/insight/problem introduced.
>
> **Response 1:**
>
> We would like to clarify that AMRe is not an incremental method but is based on a mathematical foundation. It addresses a critical issue overlooked by prior works: **incomplete and rigid multimodal masking** can cause the dominated modality to be dragged down by lazy ones. Please refer to Response 1 of Reviewer rMSX for a detailed comparison.
>
> To resolve this, we introduce:
>
> + **New Stand for Identification Laziness**: We introduce adaptive-margin modality identification by evaluating modality laziness along the dimensions of **incorrectness and uncertainty**, with $\gamma$ as a soft margin. (Please refer to our **Response 6 to Reviewer n9zw** for the instance analysis, and to our **Response 1 to Reviewer TZL5 (Dual-dimension evaluation)** for the metric analysis.)
> + **New Paradigm for Multimodal Masking input**: The core masking mechanism of AMRe is derived from a rigorous mathematical analysis of gradient coupling (App.A.1). By applying the chain rule, we show how the dominated modality suppresses lazy ones through the interaction term $\rho$ (line 576). Based on this analysis, AMRe is designed to **cut off**, **at the sample-level**, **the gradient** interference path from the dominated modality to the lazy modality, which is mathematically achieved by feature masking that sets the dominated modality's interaction term to $\rho=1$. We further propose a **epoch-level periodic restoration** mechanism, implemented through full optimization, to prevent the dominated modality from falling behind due to continuous lack of updates. The detailed different restoration method analysis are provided in our **response 7 to Reviewer n9zw**.
>
> &nbsp;
>
> **Weakness 2:** marginal improvement compared to baseline methods.
>
>  **Response 2:**
>
> While AMRe shows only marginal improvements over AMST\_FULL on CREMA-D and IEMOCAP, AMST\_FULL relies on a computationally heavy dual-stream ensemble (Concat and Sum). Operating on a single Concat framework, AMRe **halves both parameters and training time**, yet consistently outperforms AMST\_FULL across all datasets, achieving a 1.47\% higher average performance.
>
> &nbsp;
>
> **Weakness 3:** high-modality scenarios ($>$ 5 modalities).
>
> **Response 3:**
>
> Our method can handle high-modality scenarios without losing generality. We would like to take the experiments on high-modality scenarios if the relevant benchmarks are provided.
>
> &nbsp;
>
> **Weakness 4:** unimodal performances.
>
> **Response 4:**
> Thanks to the suggestions, we conducted unimodal experiments as shown in **Table 1**, and the full results table can be found at https://anonymous.4open.science/r/Rebuttal-3D2C.  The experimental results clearly show the advantage of the proposed method.
>
> **Table 1**. Results(%) of unimodal performance.
>
> |   Method   |   AVE—V   |   AVE—A   |  MVSA-I   |  MVSA-T   |      |
> | :--------: | :-------: | :-------: | :-------: | :-------: | :--: |
> |   Concat   |   37.56   |   64.68   |   59.67   |   71.93   |      |
> |   Random   |   37.56   |   62.94   |   59.04   |   71.72   |      |
> |    OGM     |   38.64   |   62.83   |   59.67   |   71.93   |      |
> |    OPM     |   39.30   |   65.67   |   57.92   |   72.35   |      |
> |    MLA     |   42.51   |   66.85   |   60.09   |   71.13   |      |
> |  Resample  |   38.06   |   66.42   | **61.95** |   72.56   |      |
> |   Remix    |   37.31   |   63.95   |   61.33   |   70.89   |      |
> | AMST_JOINT |   36.66   |   64.41   |   56.76   |   68.43   |      |
> |    AMRe    | **42.54** | **67.66** |   59.88   | **72.77** |      |
>
>
>
> &nbsp;
>
> **Weakness 5:** large-scale models as backbone.
>
> **Response 5:**
> In billion-level VLMs, gradient suppression remains a scale-invariant issue due to softmax probabilities (Eq.5). In the new study on MokA, the author also pointed out that **modal laziness still exists in VLM [1]**.
>
> > [1] Wei, Y., Miao, Y., Zhou, D., and Hu, D. Moka: Multimodal low-rank adaptation for mllms. arXiv preprint arXiv:2506.05191.
>
> &nbsp;
>
> **Weakness 6:** random masking experiments.
>
> **Response 6:** We have provided additional ablation experiments evaluating random modality masking. For the detailed results and analysis, please refer to  our **Response 1 (Deterministic Intervention) of Reviewer rMSW**.

---

> > ### Author Rebuttal · Reviewer_PKBW · 2026-04-03
> >
> > Thanks for the response. I have several follow-up questions.
> >
> > + What aspect of the method scales well with more modalities, both computationally and algorithmically?
> >
> > + Thank you for adding unimodal results. Could you clarify how these unimodal numbers are obtained? Are they measured from separately trained unimodal models?
> >
> > + If modality laziness still exists in large VLMs, would AMRe be directly applicable there, or would architectural/training differences require substantial modification?

---

> > > ### Author Response · Authors · 2026-04-04
> > >
> > > **Q 1:** What aspect of the method scales well with more modalities, both computationally and algorithmically?
> > >
> > > **Response 1:**
> > >
> > > We analyze the scalability of AMRe from both algorithmic and computational perspectives:
> > >
> > > + **Algorithmically Scalable:** At the algorithmic design level, our framework imposes no intrinsic restrictions on the number of input modalities. Because our dual-dimension evaluation metrics (incorrectness and uncertainty) are computed independently for each modality's classification head, the logic scales naturally.  For detailed derivation steps, please refer to our **second-round response to Reviewer n9zw.**
> > >
> > > + **Computationally Scalable:** From a computational perspective, AMRe is lightweight. As analyzed in detail in **Appendix D.1** (Lines 804--856), the additional time complexity introduced by evaluating a single modality is strictly bounded by $\mathcal{O}(C) + \mathcal{O}(D)$, where $C$ denotes the number of classes and $D$ denotes the dimensionality of the input features. Therefore, when extending the framework to $M$ modalities, the total additional overhead scales only linearly as $\mathcal{O}(M \times (C + D))$.
> > >
> > >   AMRe **relies on simple sample-level operations**, it completely avoids the heavy cross-modal matrix multiplications or gradient norm computations commonly used in other methods. As shown in **Table 1**, even when extending the input to three modalities (IEMOCAP), the additional training time introduced by AMRe is only about 1\% per epoch compared with the feature concatenation (Concat) baseline. This strongly demonstrates that **scaling to more modalities does not impose significant computational burden** on our method.
> > >
> > > **Table 1.** Comparison of training time per epoch of different datasets.
> > >
> > > | Method | CREMA-D |  AVE  | MVSA  | IEMOCAP |
> > > | :----: | :-----: | :---: | :---: | :-----: |
> > > | Concat | 1.00 X   | 1.00 X | 1.00 X | 1.00 X   |
> > > | AMRe   | 1.01 X   | 1.01 X | 1.00 X | 1.01 X   |
> > >
> > > &nbsp;
> > >
> > > **Q 2:**  Could you clarify how these unimodal numbers are obtained? Are they measured from separately trained unimodal models?
> > >
> > > **Response 2:**
> > >
> > > To clarify, the unimodal results in our previous rebuttal were obtained using the **full multimodal inputs** while evaluating the predictions exclusively from one modality-specific head. We have now evaluated **models trained solely on single-modality inputs**,  as shown in **Table 2**.
> > >
> > > The unimodal trained within our AMRe framework achieve performance highly comparable to the separately trained unimodal baselines (Single Input), and in some cases even surpass them. This confirms that AMRe ensures sufficient optimization for each modality, ultimately leading to the best joint performance.
> > >
> > > **Table 2.**  Result(\%) of unimodal performance across four datasets.
> > > |  Training Type   |    Method    |   CREMAD   |    AVE     |    MVSA    |  IEMOCAP   |
> > > | :--------------: | :----------: | :--------: | :--------: | :--------: | :--------: |
> > > | **Single Input** |    Audio     |   66.26%   |   66.42%   |     -      |   46.30%   |
> > > |                  |    Visual    |   70.03%   |   44.53%   |   60.50%   |   39.50%   |
> > > |                  |     Text     |     -      |     -      |   72.56%   |   62.57%   |
> > > |  **Full Input**  |   AMRe(A)    |   66.13%   |   67.66%   |     -      |   50.59%   |
> > > |                  |   AMRe(V)    |   66.13%   |   42.54%   |   59.88%   |   34.17%   |
> > > |                  |   AMRe(T)    |     -      |     -      |   72.77%   |   61.69%   |
> > > |                  | AMRe(Fusion) | **80.78%** | **76.37%** | **75.68%** | **68.93%** |
> > >
> > > &nbsp;
> > >
> > > **Q 3:**  If modality laziness still exists in large VLMs, would AMRe be directly applicable there, or would architectural/training differences require substantial modification?
> > >
> > > **Response 3:**
> > >
> > > AMRe is transferable to VLMs, but whether it can be directly applied depends largely on the VLMs architecture, especially on whether explicit unimodal classification representations are available (e.g., classification heads or [CLS] tokens):
> > >
> > > + **Without explicit signals**: **AMRe requires minor adaptation.** During training, an auxiliary unimodal classification branch (e.g., a lightweight classification head or [CLS] token) can be introduced to estimate the performance of each modality independently. After identifying the dominated and lazy modalities, the adaptive masking operation should be applied before multimodal fusion module, so as to prevent the dominated modality from suppressing lazy modalities during complex crossmodal interactions. For a discussion of interference in the parameterized fusion stage, please refer to Appendix B.1(lines  597-659).
> > > + **With explicit signals**:  **AMRe can be applied directly**. Once the performance of each modality is obtained, AMRe can perform dominated or lazy modality identification and apply adaptive masking before the final classification stage, thereby blocking the interference from dominated modality to lazy  modalities.

---

### Decision · Program_Chairs · 2026-04-30

**Decision:**

Reject

**Comment:**

This paper addresses the problem of modality imbalance (claimed modality laziness) in multimodal learning and proposes an adaptive-margin reweighting (AMRe) strategy combined with masking-based optimization. All reviewers acknowledge the importance of improving modality utilization in multimodal systems. However, several substantive concerns remain regarding the novelty, technical improvements, and empirical validation of the proposed method.

Generally, from the overall score evaluation, this paper falls into a borderline case with an average 3.5 score. **However, a key point of consensus among all the Reviewers PKBW (Partially resolved while given a second round rebuttal), rMsX (Fully resolved, appropriate weak reject), n9zw (raised the score to 4, but remaining significant concern about the originality of the work) and TZL5 (good) is that the technical contribution appears incremental.** The proposed method largely combines existing components, incorrectness (cross-entropy), uncertainty (KL divergence), and masking strategies, failing to introduce new components. In particular, Reviewer TZL5 questions whether the adaptive-margin mechanism offers meaningful novelty (dual dimensions with cross-entropy and KL divergence)  beyond a weighted combination of known metrics, and Reviewer PKBW similarly notes the heuristic nature of the approach. Although the rebuttal clarifies some design choices, concerns about originality remain only partially resolved.

Another issue is the lack of rigorous analysis of the method’s behavior and scalability. Reviewer PKBW raises important questions about how the method scales to scenarios with a larger number of modalities and whether it remains effective for modern large-scale vision-language models. These points were not fully addressed in the rebuttal. Additionally, Reviewer n9zw  highlights an important limitation: the method may conflate genuinely noisy or uninformative modalities with lazy ones, which could lead to suboptimal or even harmful training dynamics. While the authors provided clarifications, this concern still weakens the general applicability of the approach.

For empirical study, reviewers appreciate the added experiments (e.g., unimodal results), but several gaps remain. There is insufficient comparison with simpler baselines such as random modality masking (Reviewer PKBW). Furthermore, the method appears sensitive to hyperparameters, with different datasets requiring substantially different settings. In some cases, parts of the proposed formulation are effectively disabled to achieve the best performance. This raises concerns about robustness and reproducibility, as also noted by Reviewer rMsX.

The overall reviewer scores improved after rebuttal (two reviewers increased to 4), yet one reviewer further pointed out the novelty issue in the discussion. However, even after rebuttal, concerns about novelty and methodological clarity persist, and these issues matter in the paper contribution. We have to reject this paper.